# Fast Graph Sharpness-Aware Minimization for Enhancing and Accelerating Few-Shot Node Classification

**Yihong Luo**[1,2*] **& Yuhan Chen**[3*],

**Siya Qiu**[1,2]**, Yiwei Wang**[4,5]**, Chen Zhang**[6]**, Yan Zhou**[6]**, Xiaochun Cao**[7†]**, Jing Tang**[1,2†]

[1] The Hong Kong University of Science and Technology
[2] The Hong Kong University of Science and Technology (Guangzhou)
[3] School of Computer Science and Engineering, Sun Yat-sen University
[4] University of California, Merced [5] University of California, Los Angeles
[6] Createlink Technology
[7] School of Cyber Science and Technology, Shenzhen Campus of Sun Yat-sen University

## Abstract

Graph Neural Networks (GNNs) have shown superior performance in node classification. However, GNNs perform poorly in the Few-Shot Node Classification (FSNC) task that requires robust generalization to make accurate predictions for unseen classes with limited labels. To tackle the challenge, we propose the integration of Sharpness-Aware Minimization (SAM)—a technique designed to enhance model generalization by finding a flat minimum of the loss landscape—into GNN training. The standard SAM approach, however, consists of two forward-backward steps in each training iteration, doubling the computational cost compared to the base optimizer (e.g., Adam). To mitigate this drawback, we introduce a novel algorithm, Fast Graph Sharpness-Aware Minimization (FGSAM), that integrates the rapid training of Multi-Layer Perceptrons (MLPs) with the superior performance of GNNs. Specifically, we utilize GNNs for parameter perturbation while employing MLPs to minimize the perturbed loss so that we can find a flat minimum with good generalization more efficiently. Moreover, our method reutilizes the gradient from the perturbation phase to incorporate graph topology into the minimization process at almost zero additional cost. To further enhance training efficiency, we develop FGSAM+ that executes exact perturbations periodically. Extensive experiments demonstrate that our proposed algorithm outperforms the standard SAM with lower computational costs in FSNC tasks. In particular, our FGSAM+ as a SAM variant offers a faster optimization than the base optimizer in most cases. In addition to FSNC, our proposed methods also demonstrate competitive performance in the standard node classification task for heterophilic graphs, highlighting the broad applicability. The code is available at https://github.com/draym28/FGSAM_NeurIPS24

## 1 Introduction

Graph Neural Networks (GNNs) have received significant interest in recent years due to their powerful ability in various graph learning tasks, e.g., node classification. Numerous GNNs have been developed accordingly [20, 35, 6, 15]. Despite their successes, GNNs, like traditional neural networks, tend to be over-parameterized, often requiring extensive labeled data for training to ensure generalization. However, in real-world networks, many node classes have few labeled instances, which can lead

---

*Equal Contribution
†Corresponding Author: Xiaochun Cao and Jing Tang.

38th Conference on Neural Information Processing Systems (NeurIPS 2024).

to GNNs overfitting, resulting in poor generalization in these limited labeled classes. Recently, an increasing amount of research is focusing on developing superior GNNs, e.g., Meta-GCN [42], AMM-GNN [37], GPN [8] and TENT [38], for Few-Shot Node Classification (FSNC) which aims to classify nodes from new classes with limited labelled instances.

Intuitively, training GNNs for FSNC requires robust model generalization ability for recognizing unseen classes from a small number of labelled examples. Motivated by the success of the recently proposed Sharpness-Aware Minimization (SAM) for improving models' generalization in the vision domain [12], we suggest incorporating SAM into training GNNs for addressing FSNC tasks. The core idea of SAM is to perturb the model parameters to find flat minima of the loss landscape, thereby making the model more generalizable. However, a key drawback of SAM is that it requires executing two forward-backward steps to complete one optimization step, resulting in twice the time consumption compared to general optimizers like Adam. Some works [9, 23, 10] have been proposed to accelerate SAM, but none of them are crafted for graphs, i.e., not leveraging the graph properties for accelerating SAM.

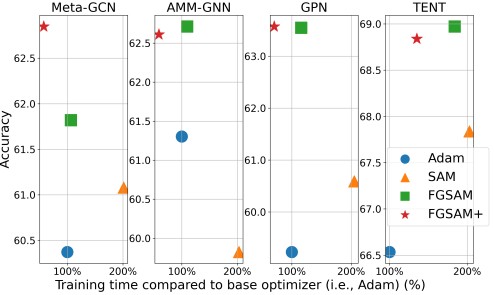

This paper mainly focuses on efficient GNN training in FSNC scenarios by leveraging SAM for improving the generalization of GNNs on unseen classes. To tackle the high training cost issue of SAM, we utilize the connection between GNNs and MLPs—GNNs discarding Message-Passing (MP) are equivalent to MLPs with faster training and worse performance in general—to accelerate training. Specifically, we propose **F**ast **G**raph **S**harpness-**A**ware **M**inimization (**FGSAM**) that uses GNNs for perturbing parameters and employs MLPs (i.e., GNNs discarding MP) to minimize perturbed training loss. This speeds up training at

Figure 1: Comparison of average accuracy and training time across datasets on different GNNs. **The closer to the top left corner, the better.**

the cost of dropping graph topology information during minimizing the perturbed loss. Interestingly, we find that the gradient computed in parameter perturbation can be reused when minimizing loss to explicitly reintroduce topology information with negligible extra cost. Moreover, we can add back MP during inference to improve performance. To further reduce the computational cost, we propose **FGSAM+** which conducts an exact FGSAM-update at every $k$ steps. As shown in Fig. 1, empirical results in FSNC tasks show that our proposed FGSAM and FGSAM+ methods outperform both Adam and SAM, and meanwhile FGSAM+ is even faster than Adam. In addition, we evaluate the proposed methods in node classification, showing strong results, especially in heterophilic graphs which are known to be challenging for GNNs [30, 7]. This indicates that our proposed methods can effectively improve the GNN's generalization capability for better performance.

The contributions of this paper can be summarized as follows.

- We study the application of SAM in FSNC tasks.
- We propose FGSAM that improves generalization in an efficient way by leveraging GNNs for sharpness-aware perturbation parameters and employing MLPs to expedite training.
- We further propose an enhanced version named FGSAM+, which conducts the actual FGSAM at every $k$ steps and approximates it in the intermediate steps.
- We demonstrate strong empirical results of the proposed methods across tasks.

## 2 Preliminary

**Graph Neural Networks.** Let $\mathcal{G} = (\mathcal{V}, \mathcal{E})$ denotes an undirected graph, $\mathcal{V} = \{v_i\}_{i=1}^n$ is the node set and $\mathcal{E} \subseteq \mathcal{V} \times \mathcal{V}$ is the edge set. $\mathbf{A} \in \mathbb{R}^{n \times n}$ is the adjacency matrix. Let $\mathbf{X} = \{\boldsymbol{x}_i\}_{i=1}^n \in \mathbb{R}^{n \times d_0}$ be the initial node feature matrix, where $d_0$ is the initial dimension, and $\mathbf{Y} = \{\boldsymbol{y}_i\}_{i=1}^n \in \mathbb{R}^{n \times C}$ denotes the ground-truth node label matrix, where $C$ denotes the number of classes and $\boldsymbol{y}_i$ is the one-hot encoding of node $v_i$'s label $y_i$. Let $\mathbf{H}^{(L)}$ be the output of the last layer of an $L$-layer GCN, the prediction probability matrix $\hat{\mathbf{Y}} = \mathrm{softmax}\left(\mathbf{H}^{(L)}\right)$ is the final output of node classification.

**Few-Shot Node Classification.** In the FSNC task, the entire set of node classes $\mathcal{C}$ can be divided into two disjoint subsets: base classes set $\mathcal{C}_{\mathrm{base}}$ and novel classes set $\mathcal{C}_{\mathrm{novel}}$, such that $\mathcal{C} = \mathcal{C}_{\mathrm{base}} \cup \mathcal{C}_{\mathrm{novel}}$ and $\mathcal{C}_{\mathrm{base}} \cap \mathcal{C}_{\mathrm{novel}} = \varnothing$. There are sufficient labeled nodes in $\mathcal{C}_{\mathrm{base}}$, while there are only a limited number of labeled nodes in $\mathcal{C}_{\mathrm{novel}}$. FSNC task aims to learn a model using the sufficient labeled nodes from

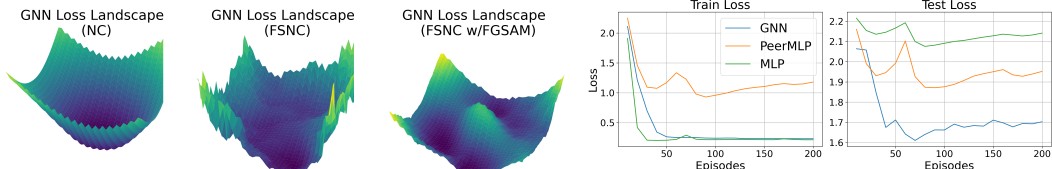

(a) Visualization of Loss Landscape                (b) Visualization of Loss Curve

Figure 2: **(a):** Loss landscape visualization of GNN across tasks and optimizers. **(b):** Loss of GNN, MLP and its PeerMLP on the test set over the training process. In these experiments, MLP and PeerMLP share the same weight space as GNN but are trained without message-passing.

$\mathcal{C}_{\text{base}}$, enabling it to accurately predict unlabeled nodes (i.e., query nodes $\mathcal{Q}$) in $\mathcal{C}_{\text{novel}}$, with limited labeled instances (i.e., support nodes $\mathcal{S}$) from $\mathcal{C}_{\text{novel}}$.

**Sharpness-Aware Minimization (SAM).** SAM [12] is an effective method to improve model's generalization. Let $\mathcal{D}_{\text{tr}} = \{(\boldsymbol{x}_i, \boldsymbol{y}_i)\}_{i=1}^{n}$ be the training dataset, following distribution $\mathcal{D}$. Given a model parameterized by $\boldsymbol{w}$ and a commonly used loss function (e.g., cross-entropy loss) $\ell$, instead of directly minimizing training loss $\mathcal{L}_{\mathcal{D}_{\text{tr}}}(\boldsymbol{w}) = \frac{1}{n}\sum_{i=1}^{n}\ell(\boldsymbol{x}_i, \boldsymbol{y}_i; \boldsymbol{w})$, SAM aims to minimize the population loss $\mathcal{L}_{\mathcal{D}}(\boldsymbol{w}) = \mathbb{E}_{(\boldsymbol{x}, \boldsymbol{y}) \sim \mathcal{D}}[\ell(\boldsymbol{x}, \boldsymbol{y}; \boldsymbol{w})]$ by minimizing the vanilla training loss as well as the loss sharpness (i.e., find parameters whose neighbors within the $\ell_p$ ball also have low training loss $\mathcal{L}_{\mathcal{D}_{\text{tr}}}$) as follows:

$$\boldsymbol{w}^* = \arg\min_{\boldsymbol{w}} \left\{ \max_{\|\boldsymbol{\epsilon}\|_p \leq \rho} \left[ \mathcal{L}_{\mathcal{D}_{\text{tr}}}(\boldsymbol{w} + \boldsymbol{\epsilon}) - \mathcal{L}_{\mathcal{D}_{\text{tr}}}(\boldsymbol{w}) \right] + \mathcal{L}_{\mathcal{D}_{\text{tr}}}(\boldsymbol{w}) + \lambda \|\boldsymbol{w}\|_2^2 \right\}$$
$$= \arg\min_{\boldsymbol{w}} \left\{ \max_{\|\boldsymbol{\epsilon}\|_p \leq \rho} \mathcal{L}_{\mathcal{D}_{\text{tr}}}(\boldsymbol{w} + \boldsymbol{\epsilon}) + \lambda \|\boldsymbol{w}\|_2^2 \right\}, \tag{1}$$

where $\rho$ is the radius of the $\ell_p$ ball, and $p \geq 0$ (usually $p = 2$). In this way, the model can converge to flat minima in loss landscape ($\boldsymbol{w}^*$), making the model more generalizable [12]. For efficiency, SAM applies first-order Taylor expansion and classical dual norm problem to obtain the approximation:

$$\hat{\boldsymbol{\epsilon}} = \rho \frac{\nabla_{\boldsymbol{w}} \mathcal{L}_{\mathcal{D}_{\text{tr}}}(\boldsymbol{w})}{\|\nabla_{\boldsymbol{w}} \mathcal{L}_{\mathcal{D}_{\text{tr}}}(\boldsymbol{w})\|} \approx \arg\max_{\|\boldsymbol{\epsilon}\|_p \leq \rho} \mathcal{L}_{\mathcal{D}_{\text{tr}}}(\boldsymbol{w} + \boldsymbol{\epsilon}). \tag{2}$$

Finally, SAM computes the gradient w.r.t. perturbed model $\boldsymbol{w} + \hat{\boldsymbol{\epsilon}}$ for update $\boldsymbol{w}$ in Eq. (1):

$$\nabla_{\boldsymbol{w}} \max_{\|\boldsymbol{\epsilon}\|_p \leq \rho} \mathcal{L}_{\mathcal{D}_{\text{tr}}}(\boldsymbol{w} + \boldsymbol{\epsilon}) \approx \nabla_{\boldsymbol{w}} \mathcal{L}_{\mathcal{D}_{\text{tr}}}(\boldsymbol{w} + \hat{\boldsymbol{\epsilon}}) \approx \nabla_{\boldsymbol{w}} \mathcal{L}_{\mathcal{D}_{\text{tr}}}(\boldsymbol{w})|_{\boldsymbol{w} + \hat{\boldsymbol{\epsilon}}}. \tag{3}$$

**Additional Related Works.** The effectiveness of SAMs and its variants have been widely verified in computer vision area [12, 21, 9, 23, 43, 10, 1]. Specifically, LookSAM [23] speeds up the SAM by periodically conducting exact perturbation, and Sharp-MAML [1] firstly focusing on meta-learning tasks. However, there is limited work on developing SAM for graphs. WT-AWP [39] is the first SAM-like work that applied to GNN and gives a theoretical analysis of generalization bound on graphs. Compared to these works, our proposed FGSAM is crafted for graphs by its unique property, enabling the *first SAM-like algorithm that can be faster than the base optimizer.* Our work also shares some similarities with existing works [17, 40] that explore the connection between GNNs and MLPs. However, they attributed the claim that introducing MP to MLP can improve performance during evaluation to the powerful generalization ability of MP. In contrast, we prove that for the linear case with synthetic graphs, whether there is MP or not, both will converge to the same optimal solution, taking a solid step toward understanding the underlying reasons.

## 3 Methodology

In this section, we propose Fast Graph Sharpness-Aware Minimization (FGSAM), an efficient version of SAM for GNNs, aiming to reduce the training time when using SAM in FSNC tasks while improving model's generalization.

### 3.1 Motivating Analysis

SAMs are a series of new general training scheme used to improve the model's generalization, thus it is intuitive to use SAM in FSNC tasks. However, there is no work studying how to apply SAM to FSNC tasks. So our first question is: **Q1: Can SAM benefit few-shot node classification tasks?**

Table 1: Time consumption of 200 episodes training (sec.) of baseline w/ and w/o MP (only consider feed-forward and -backward).

| Bseline | Backbone | CoraFull | | DBLP | | ogbn-A | |
| | | 5N3K | 10N3K | 5N3K | 10N3K | 5N3K | 10N3K |
| --- | --- | --- | --- | --- | --- | --- | --- |
| Meta-GCN | GNN | 9.56 | 9.38 | 17.61 | 17.50 | 41.09 | 40.96 |
| | PeerMLP | 1.11 | 1.17 | 1.35 | 1.54 | 1.02 | 1.17 |

A key property of FSNC is that the GNNs need to be generalized to unseen classes (i.e., novel classes), and the GNNs often converge to a relatively low loss on the training set, but the final performance depends on the GNNs' generalization ability. To demonstrate this intuitively, we plot the GNN's loss landscape of novel classes under the FSNC setting and of the test set under the NC setting (Fig. 2a), following previous work [22]. The loss landscape of GNN under the FSNC setting is sharp and not smooth, with many local minima, in contrast to the flat and smooth loss landscape of GNN under the NC setting. This to some extent indicates that the FSNC setting poses a greater challenge to GNNs, which is consistent with our prior knowledge. Hence, applying SAM-like techniques can intuitively improve the generalization of GNN and enhance its performance.

However, another problem arises: training GNN on FSNC is already slow, and the core drawback of SAM is that it requires twice the training cost compared to Adam or SGD. **Q2: Can we find a way to reduce the SAM training cost based on GNN properties?**

It is well known that the training speed of GNNs is slower than MLPs, mainly due to the notorious MP that causes significant time consumption, yet MP is essential for improving GNN performance. Removing the MP from GNNs $f_{\text{gnn}}(\{\mathbf{X}, \mathbf{A}\}; \boldsymbol{w})$ turns them into MLPs $f_{\text{mlp}}(\mathbf{X}; \boldsymbol{w})$, which is an intriguing connection. As shown in Tab. 1 and Fig. 2b, MLPs without the burden of MP demonstrate a substantial training time advantage under the same settings as GNNs and can achieve nearly the same performance as GNNs on the training set, however, they perform significantly worse on the test set, revealing their poor generalization performance.

Inspired by previous work [17], it is appealing to remove MP during training, but reintroduce it in inference (**PeerMLP**). Although reintroducing MP after training can improve the performance, it still cannot surpass GNNs' (Fig. 2b). This may be because of the lack of graph topology information in training. Hence, we propose minimizing training loss on PeerMLPs but minimizing the sharpness according to GNNs, implicitly incorporating the graph topology information in training. This allows the model to quickly converge to the vicinity of local minima and further converge to flat GNN local minima through a GNN's sharpness-aware approach. By doing so, we not only introduce SAM to enhance the model's generalization ability and the information w.r.t graph topology but also leverage the intriguing connection between MLPs and GNNs to improve training speed.

### 3.2 FGSAM

We elaborate our proposed method **Fast Graph Sharpness-Aware Minimization (FGSAM)**. For the ease of reference, Fig. 3a visualizes the framework of FGSAM, so does to its enhanced version FGSAM+. There are two forward-backward steps in the FGSAM-update.

**Step 1: Graph sharpness-aware perturbation.** The first forward-backward step is served for computing the maximum perturbation $\hat{\epsilon}$ (Eq. (2)), where we propose to perturb parameters with MP (GNN), i.e.,

$$\hat{\epsilon} = \rho \frac{\boldsymbol{g}^{\text{gnn}}}{\|\boldsymbol{g}^{\text{gnn}}\|} = \rho \frac{\nabla_{\boldsymbol{w}} \mathcal{L}_{\mathcal{G}}(\boldsymbol{w}; f_{\text{gnn}})}{\|\nabla_{\boldsymbol{w}} \mathcal{L}_{\mathcal{G}}(\boldsymbol{w}; f_{\text{gnn}})\|} = \rho \frac{\nabla_{\boldsymbol{w}} \mathcal{L}(f_{\text{gnn}}(\mathcal{G}; \boldsymbol{w}), \mathbf{Y})}{\|\nabla_{\boldsymbol{w}} \mathcal{L}(f_{\text{gnn}}(\mathcal{G}; \boldsymbol{w}), \mathbf{Y})\|} \tag{4}$$

**Step 2: Minimizing perturbed loss.** We propose to minimize the perturbed loss by removing the MP (PeerMLP) to speed up training, i.e.,

$$\boldsymbol{w}^* = \arg\min_{\boldsymbol{w}} \mathcal{L}_{\mathbf{X}}(\boldsymbol{w} + \hat{\epsilon}; f_{\text{mlp}}) = \arg\min_{\boldsymbol{w}} \mathcal{L}(f_{\text{mlp}}(\mathbf{X}; \boldsymbol{w} + \hat{\epsilon}), \mathbf{Y})$$
$$= \arg\min_{\boldsymbol{w}} \mathcal{L}(f_{\text{gnn}}(\hat{\mathcal{G}} = \{\mathbf{X}, \mathbf{I}\}; \boldsymbol{w} + \hat{\epsilon}), \mathbf{Y}). \tag{5}$$

It is clear that minimizing the loss on PeerMLPs is equivalent to minimizing the loss on GNNs ignoring the topology information. As demonstrated in Sec. 3.1, intuitively the proposed approach can make model convergence near the local minima easily due to the connection between MLPs and GNNs, and perturbing parameters with MP can find the good flat minima of GNNs (see Fig. 2a).

**Reintroducing Graph Topology in Minimization with Free Lunch.** While reintroducing the MP in evaluation can improve performance, its absence during the minimization process may result in

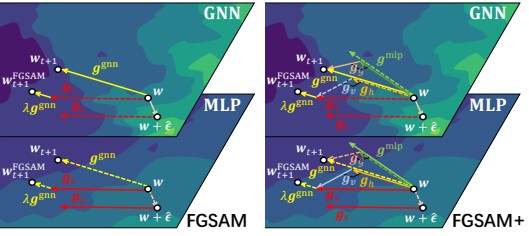 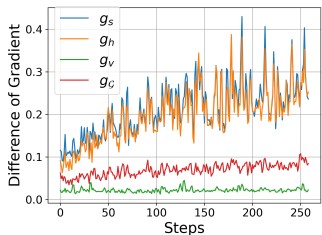

(a) Visualization of FGSAM and FGSAM+        (b) Gradient difference curves

Figure 3: **Left (a):** The solid line indicates that the gradient is computed on the corresponding model, while the dashed line indicates the opposite. **Right (b):** The difference of gradients (i.e., $\|\boldsymbol{g}_{t+1} - \boldsymbol{g}_t\|_2$). It can be seen that $\boldsymbol{g}_v$ and $\boldsymbol{g}_{\mathcal{G}}$ change much slower than $\boldsymbol{g}_s$ and $\boldsymbol{g}_h$ across the training process, thus can be reused in the intermediate steps.

sub-optimal results. Incorporating MP directly into the minimization is computationally expensive, leading us to employ MLP to minimize the perturbed loss. Fortuitously, the gradient w.r.t. MP is computed during the perturbation step, offering an opportunity for computational savings. We propose to capitalize on the already available gradient information from the first step by reusing it in the optimization procedure, as formalized in the following optimization target:

$$\boldsymbol{w}^* = \arg\min_{\boldsymbol{w}} \left\{ \lambda \times \mathcal{L}_{\mathcal{G}}(\boldsymbol{w}; f_{\text{gnn}}) + \mathcal{L}_{\mathbf{X}}(\boldsymbol{w} + \hat{\boldsymbol{\epsilon}}; f_{\text{mlp}}) \right\}, \quad \lambda \geq 0. \tag{6}$$

This formulation implies that the computational cost of involving MP in the optimization is mitigated since the forward and backward passes are precomputed in the initial step. Thus, we effectively integrate graph topology into the minimization process almost without incurring additional computational expense, akin to receiving a *free lunch*. See detailed **FGSAM** in Algorithm 1.

**Adaptation to MAML Models.** Model-Agnostic Meta-Learning (MAML) [11] is widely used in FSNC tasks [8, 38], involving two separate update steps in one MAML-update: i) pre-training for learning task-relevant knowledge, and ii) meta-update for task-irrelevant update. This is different from standard gradient descent. Hence for integrating the FGSAM into the MAML models, we propose treating the MAML-update process as a single entity, and applying the FGSAM-update only once simplifies the implementation. This contrasts with the Sharp-MAML [1], where the SAM-update is applied separately in the two stages.

### 3.3 FGSAM+

Although the training time of FGSAM can be largely faster than naïve SAM by ignoring the MP in minimizing perturbed loss, it still requires a full forward-backward step of GNN, which makes our approach need an extra computation cost for a forward-backward step of PeerMLP, compared to the base optimizer.

Fortunately, the forward-backward step of GNN is mainly for perturbing parameters in FGSAM, thus we can further reduce the training time while maintaining performance, by employing FGSAM-update at every $k$ step (i.e., perturb parameters at every $k$ step) and reusing the preserved gradients from parameters perturbation into the intermediate steps [23]. Eq. (3) can be rewritten as:

$$\nabla_{\boldsymbol{w}} \mathcal{L}_{\mathcal{D}_{\text{tr}}}(\boldsymbol{w})|_{\boldsymbol{w}+\hat{\boldsymbol{\epsilon}}} \approx \nabla_{\boldsymbol{w}} \mathcal{L}_{\mathcal{D}_{\text{tr}}}(\boldsymbol{w} + \hat{\boldsymbol{\epsilon}}) \approx \nabla_{\boldsymbol{w}} \Big[ \mathcal{L}_{\mathcal{D}_{\text{tr}}}(\boldsymbol{w}) + \rho \| \nabla_{\boldsymbol{w}} \mathcal{L}_{\mathcal{D}_{\text{tr}}}(\boldsymbol{w}) \| \Big]. \tag{7}$$

In this way, SAM-gradient $\boldsymbol{g}_s$ is composed by the vanilla gradient $\nabla_{\boldsymbol{w}} \mathcal{L}_{\mathcal{D}_{\text{tr}}}(\boldsymbol{w})$ and the gradient of the $\ell_2$-norm of vanilla gradient $\nabla_{\boldsymbol{w}} \| \nabla_{\boldsymbol{w}} \mathcal{L}_{\mathcal{D}_{\text{tr}}}(\boldsymbol{w}) \|$.

This suggests that SAM-gradient $\boldsymbol{g}_s = \nabla_{\boldsymbol{w}} \mathcal{L}_{\mathcal{D}_{\text{tr}}}(\boldsymbol{w})|_{\boldsymbol{w}+\hat{\boldsymbol{\epsilon}}}$ can be divided into two orthogonal parts [23]: $\boldsymbol{g}_h$ (in the direction of vanilla gradient $\boldsymbol{g} = \nabla_{\boldsymbol{w}} \mathcal{L}_{\mathcal{D}_{\text{tr}}}(\boldsymbol{w})$ ) is used to minimize the loss value, and flatness-gradient $\boldsymbol{g}_v$ is used to adjust the updates towards a flat region. So $\boldsymbol{g}_h$ and $\boldsymbol{g}_v$ can be easily obtained if $\boldsymbol{g}_s$ and $\boldsymbol{g}$ are given:

$$\boldsymbol{g}_h = \|\boldsymbol{g}_s\| \cos\theta \frac{\boldsymbol{g}}{\|\boldsymbol{g}\|} = \|\boldsymbol{g}_s\| \frac{\boldsymbol{g}_s \cdot \boldsymbol{g}}{\|\boldsymbol{g}_s\| \|\boldsymbol{g}\|} \frac{\boldsymbol{g}}{\|\boldsymbol{g}\|}, \quad \boldsymbol{g}_v = \boldsymbol{g}_s - \boldsymbol{g}_h, \tag{8}$$

where $\theta$ is the angle between $\boldsymbol{g}_s$ and $\boldsymbol{g}_h$. As illustrated in [23], $\boldsymbol{g}_v$ changes much slower than $\boldsymbol{g}_s$ and $\boldsymbol{g}_h$, thus we can compute and preserve $\boldsymbol{g}_v$ at every $k$ steps, and reuse it to approximate $\boldsymbol{g}_s$ in intermediate steps.

However, in our case, there exists a clear gap between the model used for perturbing (GNN) and for minimizing (PeerMLP). This is different from the approach in [23], which uses the same model for both. Thus we use an extra PeerMLP forward-backward step to get another $g^{\mathrm{mlp}}$ for computing $g_v$ to reduce the gap:

$$g^{\mathrm{mlp}} = \nabla_{\boldsymbol{w}} \mathcal{L}(f_{\mathrm{mlp}}(\mathbf{X}; \boldsymbol{w})) = \nabla_{\boldsymbol{w}} \mathcal{L}(\boldsymbol{w}; f_{\mathrm{mlp}}), \quad g_s = \nabla_{\boldsymbol{w}} \mathcal{L}(\boldsymbol{w}; f_{\mathrm{mlp}})|_{\boldsymbol{w}+\hat{\epsilon}}. \tag{9}$$

Note that the $\hat{\epsilon}$ is obtained by perturbing parameters with MP Eq. (4), $g_s$ and $g^{\mathrm{mlp}}$ are obtained without MP, thus there still exists a gap.

Moreover, since we reintroduce graph topology (Eq. (6)) in minimization, we propose to further use the extra PeerMLP step to reuse graph topology for better performance. Specifically, we conduct the gradient w.r.t. topology information by projection as follows:

$$g_{\mathcal{G}} = g^{\mathrm{gnn}} - \|g^{\mathrm{gnn}}\| \cos(\theta') \frac{g^{\mathrm{mlp}}}{\|g^{\mathrm{mlp}}\|}, \tag{10}$$

where $\theta'$ is the angle between $g^{\mathrm{gnn}}$ and $g^{\mathrm{mlp}}$. This can be reused in a similar way as $g_v$ when approximating FGSAM-update in the intermediate steps. We further conduct experiments to verify whether the $g_{\mathcal{G}}$ and $g_v$ will change slowly so that they can be reused for speed up in our approach. We plot the change of $g_s$, $g_h$, $g_v$ and $g_{\mathcal{G}}$ (Fig. 3b) and the results show that the projected gradient both $g_v$ and $g_{\mathcal{G}}$ on parameters perturbed with MP shows a much more stable pattern and slower changes than $g_s$ and $g_h$, indicating the feasibility of updating $g_v$ and $g_{\mathcal{G}}$ every $k$ steps and reusing it for the intermediate steps. We present the detailed **FGSAM+** in Algorithm 1 in Appendix B.

Since we need an extra PeerMLP forward-backward step at every $k$ step, the overall computation cost of our approach, FGSAM+, will be $\frac{1}{k} \times$ the computation cost of GNNs plus $(1 + \frac{1}{k}) \times$ the computation cost of MLPs on average.

## 4 Analysis of Toy Case

In this section, we employ the Contextual Stochastic Block Model (CSBM) to analyze why minimizing perturbed training loss without MP can work to some extent, which is the underlying mechanism of FGSAM. The CSBM has been widely used to analyze of the properties of GNN [27, 26].

Specifically, we focus on a CSBM model that contains $K$ distinct classes $c_1, c_2, \ldots, c_K$. The nodes within the resulting graphs are grouped into $n$ non-overlapping sets $C_1, C_2, \ldots, C_K$, each set representing one of the $K$ classes. The generation of edges is governed by a probability $p$ within the same class and a probability $q$ between different classes. For any given node $i$, we sample its initial features $\boldsymbol{x}_i \in \mathbb{R}^l$ from a Gaussian distribution denoted by $\boldsymbol{x}_i \sim \mathcal{N}(\boldsymbol{\mu}, \mathbf{I})$, where the mean $\boldsymbol{\mu} = \boldsymbol{\mu}_k \in \mathbb{R}^l$ corresponds to node $i$ belonging to set $C_K$, and $k$ is an element of $\{1, 2, \ldots, K\}$. Furthermore, the condition $\|\boldsymbol{\mu}_i - \boldsymbol{\mu}_j\|_2 = D$ holds true for all $i, j$ belonging to $\{1, 2, \ldots, K\}$, with $D$ being a positive constant. Graphs that arise from this specified CSBM model are referred to as $K$-classes CSBM. After applying a MP operation, the resultant features for node $i$ are denoted by $\boldsymbol{h}_i$.

The neighborhood label distribution $\mathcal{D}_i$ of node $i$ is a K-dimensions vector, where $\mathcal{D}_i[j] = \mathbb{I}(i \in C_j)p + (1 - \mathbb{I}(i \in C_j))q$. Based on the neighborhood label distribution, consider the MP operation as $\boldsymbol{h}_i = \frac{1}{deg(i)} \sum_{j \in \mathcal{N}(i)} \boldsymbol{x}_i$, we have: $\boldsymbol{h}_i \sim \mathcal{N}\left(\frac{(p-q)\boldsymbol{\mu}_k + qK\bar{\boldsymbol{\mu}}}{p+(K-1)q}, \frac{\mathbf{I}}{deg(i)}\right)$, where $i \in C_k$ and $\bar{\boldsymbol{\mu}} = \frac{\sum_{j=1}^{K} \boldsymbol{\mu}_j}{K}$. Based on the distribution of $\boldsymbol{h}_i$ and $\boldsymbol{x}_i$, we can obtain following theorem:

**Theorem 4.1** (The effectiveness of removing MP in minimization). *Consider a K-classes CSBM, the optimal linear classifiers for both original features $\boldsymbol{x}_i$ and filtered features $\boldsymbol{h}_i$ are the same.*

Detailed proof is in Appendix C. The theorem tells us that under the linear case, whether the MP layer is used or not, the optimal decision bound is the same. Hence, this encourages us to learn the weight of transformation layers without MP to speed up training. However, the real graph is more complex and we do not use a linear classifier, thus we propose to perform the graph sharpness-aware perturbation which implicitly involves the information of neighbors.

## 5 Experiments

We verify the effectiveness of our proposed FGSAM and FGSAM+ in this section. We first conduct experiments to demonstrate that our proposed algorithms achieve better performance compared to SAM which requires twice the training time. Then we show that our proposed algorithms can achieve faster training speed compared to base optimizers (e.g., Adam). Next, we also conduct extra studies and an extra task to show the robustness and potential applications of our proposed algorithms.

Table 2: Accuracy and Time consumption on the baseline with different optimizer. The best and the runner-up are denoted as boldface and underlined, respectively. '5N3K' denotes 5-way 3-shot setting. Time consumption of 200 episodes of training (sec., only consider forward-backward) is also shown.

| Setting | Corafull | | | | Avg | | DBLP | | | | Avg | | ogbn-arXiv | | | | Avg | |
|---|---|---|---|---|---|---|---|---|---|---|---|---|---|---|---|---|---|---|
| | 5N3K | 5N5K | 10N3K | 10N5K | acc | time | 5N3K | 5N5K | 10N3K | 10N5K | acc | time | 5N3K | 5N5K | 10N3K | 10N5K | acc | time |
| *MAML models* | | | | | | | | | | | | | | | | | | |
| **Meta-GCN** | 70.25 | 77.00 | 51.19 | 58.85 | 64.32 | 9.48 | 82.60 | 85.20 | 65.96 | 70.85 | 76.15 | 17.57 | 49.32 | 54.37 | 30.68 | 28.20 | 40.64 | 40.99 |
| w/ SAM | 70.23 | 75.82 | 54.77 | 58.18 | 64.75 | 19.03 | 82.50 | 85.04 | 68.31 | 71.22 | 76.77 | 35.30 | **54.80** | 55.19 | 25.10 | 31.79 | 41.72 | 82.54 |
| w/ FGSAM | 70.97 | 77.64 | 55.53 | 59.30 | 65.86 | 10.83 | **82.66** | **85.26** | **69.22** | 71.80 | **77.24** | 19.15 | 52.45 | 57.05 | 28.92 | 31.03 | 42.36 | 42.48 |
| w/ FGSAM+ | **71.54** | **78.97** | **58.73** | **61.61** | **67.71** | **6.51** | 82.40 | 84.24 | 68.97 | **72.18** | 76.95 | **10.62** | 52.98 | **58.08** | **31.09** | **33.38** | **43.88** | 22.11 |
| **AMM-GNN** | **72.92** | **80.44** | 57.58 | 57.29 | 67.06 | 15.00 | 81.02 | 83.48 | 66.40 | 71.31 | 75.55 | 26.73 | **51.95** | **57.79** | 28.71 | 26.74 | 41.30 | 42.33 |
| w/ SAM | 68.47 | 74.10 | 52.43 | 57.94 | 63.24 | 30.83 | 80.54 | 83.45 | 66.29 | 71.50 | 75.45 | 54.76 | 49.42 | 50.75 | 30.57 | 32.42 | 40.79 | 84.93 |
| w/ FGSAM | 71.67 | 77.72 | **60.15** | 62.11 | 67.91 | 17.60 | **84.01** | **85.32** | 67.12 | **71.70** | **77.04** | 30.16 | 48.69 | 55.89 | **35.59** | 32.57 | **43.19** | 44.41 |
| w/ FGSAM+ | 72.79 | 79.18 | 59.59 | **62.61** | **68.54** | **10.00** | 81.24 | 85.07 | **70.37** | 71.32 | 77.00 | **16.26** | 51.02 | 50.49 | 33.60 | **34.05** | 42.29 | 23.19 |
| *non-MAML models* | | | | | | | | | | | | | | | | | | |
| **GPN** | 65.23 | 65.67 | 50.48 | 51.23 | 58.15 | 1.89 | 76.05 | 75.02 | 65.41 | 64.52 | 70.25 | 3.28 | 55.35 | 57.50 | 42.72 | 41.54 | 49.28 | 7.70 |
| w/ SAM | 67.28 | 65.02 | 55.06 | 52.30 | 59.92 | 3.62 | 79.44 | 77.66 | 67.88 | 67.78 | 73.19 | 6.78 | 56.18 | **58.65** | 39.91 | 39.92 | 48.67 | 15.98 |
| w/ FGSAM | **69.54** | 69.37 | **57.85** | 56.49 | **63.31** | 2.33 | **80.10** | 79.61 | 68.50 | 69.44 | 74.41 | 4.00 | **57.58** | 58.23 | **47.67** | **48.20** | **52.92** | 8.57 |
| w/ FGSAM+ | 69.40 | **69.96** | 57.74 | **56.10** | 63.30 | **1.83** | 80.02 | **79.69** | **68.94** | **69.51** | **74.54** | **2.56** | 57.39 | 58.04 | 46.59 | **49.49** | 52.88 | **4.66** |
| **TENT** | 71.24 | 75.49 | 57.29 | 60.35 | 66.09 | 10.88 | 80.67 | 82.74 | 69.04 | 71.79 | 76.06 | **11.36** | 60.44 | 67.34 | 47.14 | 54.88 | 57.45 | 12.90 |
| w/ SAM | 71.38 | 75.29 | 56.86 | 61.85 | 66.35 | 22.03 | 82.13 | 85.10 | 68.96 | 73.62 | 77.45 | 22.86 | 63.58 | 69.30 | 50.79 | 55.21 | 59.72 | 26.43 |
| w/ FGSAM | 71.10 | 76.72 | 57.86 | **63.71** | 67.35 | 20.28 | 82.99 | **86.13** | 70.31 | 73.41 | 78.21 | 20.95 | 63.88 | **71.15** | **53.32** | **57.08** | **61.36** | 23.40 |
| w/ FGSAM+ | **72.85** | **77.77** | **58.37** | 63.04 | **68.01** | 15.10 | **83.64** | 85.97 | **71.15** | **73.72** | **78.62** | 15.58 | **66.20** | 69.14 | 50.66 | 53.56 | 59.89 | 16.86 |

Table 3: Comparison between SAM variants regarding accuracy and time consumption (10N3K).

| Settings | | | CoraFull | | | | DBLP | | | | ogbn-arXiv | | | | Avg | |
|---|---|---|---|---|---|---|---|---|---|---|---|---|---|---|---|---|
| | | | 5N3K | | 10N3K | | 5N3K | | 10N3K | | 5N3K | | 10N3K | | | |
| Baseline | Backbone | Optimizer | acc (%) | t (s) | acc (%) | t (s) | acc (%) | t (s) | acc (%) | t (s) | acc (%) | t (s) | acc (%) | t (s) | acc (%) | t (s) |
| GPN | GNN | Adam | 65.23 | 1.84 | 50.48 | **1.87** | 76.05 | 3.26 | 65.41 | 3.29 | 55.35 | 7.67 | 42.72 | 7.67 | 59.21 | 4.27 |
| | | SAM | 67.28 | 3.68 | 55.06 | 3.55 | 79.44 | 6.76 | 67.88 | 6.76 | 56.18 | 15.90 | 39.91 | 15.95 | 60.96 | 8.77 |
| | | ESAM | 67.32 | 3.75 | 53.99 | 3.60 | 77.58 | 6.83 | 66.54 | 6.83 | 54.51 | 16.03 | 36.68 | 16.14 | 59.44 | 8.86 |
| | | LookSAM | 68.38 | 2.91 | 54.26 | 2.85 | 79.24 | 5.29 | **69.32** | 5.29 | 56.33 | 12.23 | 45.42 | 12.19 | 62.16 | 6.79 |
| | | AE-SAM | 67.48 | 2.93 | 51.27 | 2.81 | 79.84 | 5.17 | 67.23 | 5.24 | 56.43 | 12.26 | 43.51 | 12.28 | 60.96 | 6.78 |
| | | **FGSAM** | 69.54 | 2.33 | **57.85** | 2.40 | 80.10 | 3.97 | 68.50 | 4.03 | **57.58** | 8.56 | **47.67** | 8.58 | **63.54** | 4.98 |
| | | **FGSAM+** | 69.40 | 1.62 | 57.74 | 2.06 | 80.02 | 2.43 | 68.94 | 2.59 | 57.39 | 4.68 | 46.59 | 4.64 | 63.35 | 3.00 |
| | PeerMLP | Adam | 65.80 | 0.45 | 49.87 | **0.43** | 76.41 | **0.39** | 65.00 | 0.47 | 49.09 | **0.33** | 35.98 | 0.36 | 57.03 | **0.41** |
| | | SAM | **66.18** | 0.74 | **51.69** | 1.01 | **77.20** | 0.71 | **65.39** | 0.76 | **51.75** | 0.69 | **42.79** | 0.81 | **59.17** | 0.79 |

## 5.1 Experiment Settings

**Baseline.** We evaluate our proposed FGSAM and FGSAM+ on SOTA models. The existing models can be divided into two main categories: MAML and non-MAML methods. Two representative models are selected from each category, respectively, as baselines for evaluation (**Meta-GCN** [42] and **AMM-GNN** [37] for MAML models, and **GPN** [8] and **TENT** [38] for non-MAML models).

**Datasets.** We conduct evaluations on three widely used real-world benchmark node classification datasets: `CoraFull` [5], `DBLP` and `ogbn-arXiv` [18], and we use the train/val/test split as in [34] and [24]. The comprehensive statistics of datasets are shown in Tab. 5 in Appendix D.1.

**Implementation Details.** We implement our model by PyTorch [29] and conduct experiments on an RTX-3090Ti. We use Optuna [2] to search the hyper-parameters for each setting. See Appendix D.2 for detailed FSNC learning protocol.

## 5.2 Evaluation on Real-World Datasets

The results of different models across datasets are summarized in Tab. 2. All the models share a 2-layers architecture with 16 hidden channels. It can be seen that our proposed algorithms FGSAM and FGSAM+ provide better performance than Adam in most cases, and provide comparable performance with SAM. These results support our claim that FGSAM and FGSAM+ can find local minima with better generalization properties. Note that message-passing is only used in perturbing parameters, not involved in parameters update (i.e., MLPs). The results further indicate that implicitly involving graph topology in training can make PeerMLPs outperform GNNs. See Appendix D.3 for details.

## 5.3 Time Consumption

To demonstrate the training speed advantage of our proposed algorithm, we summarize the training time for different models using various optimization methods across three datasets (Tab. 2). The results indicate that our proposed algorithm FGSAM demonstrates only a slight increase in training cost compared to Adam in most cases. Furthermore, our enhanced version FGSAM+ outperforms Adam in terms of speed in the majority of scenarios. It is worth mentioning that our proposed algorithms achieve superior or comparable performance when compared to both Adam and SAM. See Appendix D.4 for detailed results.

**Limitation.** For models composed of many non-GNN components (e.g., TENT), the training time on FGSAM+ may be still longer than that on Adam, since it is hardly further reduced.

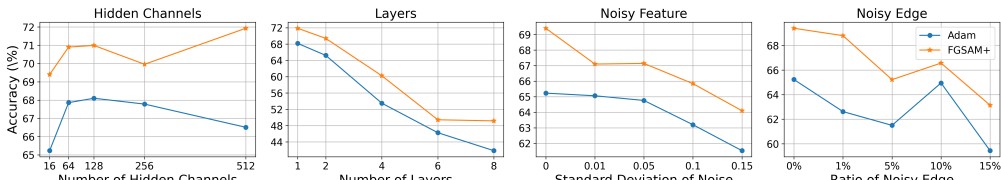

Figure 4: Performance of GPN trained by Adam and FGSAM+ with different settings. **Left:** Results with various hidden channels. **Middle Left:** Results with various model depths. **Middle Right:** Results with features perturbed by noise of varying standard deviations. **Right:** Results with edges subjected to various noise ratios.

## 5.4 Comparison of the Variants of SAM

**Training with Different Optimizer.** We compare the performance of Meta-GNN and GPN training with different variants of SAM, including original SAM, ESAM [9], LookSAM [23], AE-SAM [19], our proposed FGSAM and FGSAM+ (Tab. 3). We observe an anomalous phenomenon where ESAM, as an efficient variant of SAM, actually trains slower than SAM. This is because ESAM sorts the sample losses and selects a suitable subset at each iteration, an operation that is negligible for image tasks; however, for graph tasks, since GNNs are relatively smaller, the proportion of time consumed by the sorting step is significant, leading to an increase in training time. As shown in Tab. 3, our proposed method greatly reduces the training time, based on the relationship between GNN and MLP, while maintaining and even achieving superior performance, compared to other optimizers, indicating ours' high efficiency and effectiveness.

**The Impact of Perturbing Parameters with Message-Passing.** A key point of our work is that we perform parameter perturbation using GNNs, while PeerMLPs (i.e., without message-passing) are used to minimize the perturbed loss. This is significantly distinct from previous SAM methods which shared the same model for both parameter perturbation and loss minimization. So a natural question arises: **to what extent does our approach benefit from performing parameter perturbation using GNNs?** We thus compare our approach to PeerMLPs training with Adam and vanilla SAM. Note that message-passing would be reintroduced during validation and test. From Tab. 3, although the training time of PeerMLPs is shorter than that of GNNs, GNNs outperform their PeerMLPs in most cases. Despite that using PeerMLPs can accelerate the training of GNNs, the topology information is still very important for learning node representations. Thus our proposed FGSAM+ is a better solution, achieving a better trade-off between efficiency and performance.

## 5.5 Ablation Studies

We further verify the consistent effectiveness of our method compared to Adam across different settings regarding model implementation and graph property. Due to the computational resource restriction, all experiments here were conducted using GPN on the CoraFull with the 5-way 3-shot setting. We provide additional experiments (e.g., the effect of update interval $k$) in the Appendix E.

**The Impact of Network Structure.** Here we investigate the effect of hidden dimension and the number of layers on the performance (on the left of Fig. 4). GPN with Adam requires a higher hidden dimension (128) to achieve relatively high accuracy, whereas GPN with FGSAM+ can attain SOTA even with a small hidden dimension (16). With respect to the number of layers, GPN with FGSAM+ consistently performs better within the range of 1∼8 compared to GPN with Adam, demonstrating the effectiveness of our proposed method (middle left of Fig. 4).

**The Impact of Noisy Features and Edges.** Here we investigate the effect of randomly adding Gaussian noise to features and randomly adding edges during testing (on the middle right and the right of Fig. 4). Specifically, for noisy features, we randomly add Gaussian noise with varying standard deviations to the node features. Meanwhile, for noisy edges, we uniformly and randomly introduce additional edges into the original structure. The results show that GPN with FGSAM+ method can still achieve relatively high performance, compared to GPN with Adam. These results effectively verify the robustness of our proposed method.

## 5.6 Additional Task on Conventional Node Classfication

Our proposed FGSAM+ also has the potential to be extended to other domains. To demonstrate this, we evaluate the performance of the FGSAM+ on the standard node classification task on both homophilic and heterophilic graphs. For homophilic graphs, we utilize three well-established citation networks: `Cora`, `Citeseer`, and `Pubmed` [32, 13]. For heterophilic graphs, we include page-page

Table 4: Results on nine real-world node classification benchmark datasets: Mean accuracy (%).

| Model | Optimizer | Cora | Citeseer | Pubmed | Chameleon | Squirrel | Actor | Cornell | Texas | Wisconsin | Avg |
|---|---|---|---|---|---|---|---|---|---|---|---|
| GCN | Adam | 88.36 | 77.25 | 88.71 | 65.04 | 52.49 | 28.54 | 61.08 | 60.27 | 55.29 | 64.11 |
| | SAM | **88.42** | 77.30 | 88.79 | 65.57 | 52.51 | 28.59 | 61.89 | 62.70 | 54.51 | 64.48 |
| | **FGSAM (ours)** | 88.36 | **77.60** | **89.36** | **66.16** | **53.95** | **29.88** | 67.30 | 63.24 | **55.69** | **65.73** |
| | **FGSAM+ (ours)** | 88.32 | 77.52 | 89.13 | 64.56 | 51.14 | 29.66 | **68.11** | 61.62 | 54.71 | 64.97 |
| GraphSAGE | Adam | 87.67 | 76.09 | 89.15 | 50.33 | 37.61 | 33.74 | 78.11 | 78.38 | 84.51 | 68.40 |
| | SAM | 87.69 | 76.44 | 89.25 | 50.92 | 37.44 | 33.83 | 78.92 | 80.27 | 84.31 | 68.79 |
| | **FGSAM (ours)** | **88.36** | 77.13 | **89.75** | **51.34** | **39.12** | 34.53 | **82.43** | **81.35** | **86.47** | **70.05** |
| | **FGSAM+ (ours)** | 88.16 | **77.21** | 89.71 | 50.94 | 38.87 | **34.70** | 81.35 | 79.46 | **86.47** | 69.65 |
| GAT | Adam | 88.32 | 76.37 | 87.48 | 46.51 | 31.46 | 29.45 | 59.19 | 62.16 | 55.49 | 59.60 |
| | SAM | 88.49 | 76.78 | 87.24 | 46.82 | 31.61 | 29.49 | 59.46 | 62.16 | 55.29 | 59.70 |
| | **FGSAM (ours)** | 88.60 | 76.98 | 87.63 | 47.87 | 32.35 | 30.41 | 61.89 | **65.95** | **59.41** | **61.23** |
| | **FGSAM+ (ours)** | **88.70** | **77.10** | **87.74** | **48.07** | **32.69** | **30.60** | **62.16** | 64.86 | 58.04 | 61.11 |

networks from Wikipedia, specifically the `Chameleon` and `Squirrel` datasets [31], actor-network, namely `Actor` [30], and web pages networks, namely `Cornell`, `Texas` and `Wisconsin` [30]. See Appendix E.1 for statistics of these datasets. We use data splits (48%/32%/20%) provided by [30], and set $k = 2$ for FGSAM+. We select three representative baselines, namely the classical **GCN** [20], **GAT** [35] with learnable MP operation, and **GraphSAGE** [16] with complex MP operation, to demonstrate the effectiveness of FGSAM and FGSAM+.

As shown in Tab. 4, both FGSAM and FGSAM+ generally outperform Adam and SAM across base models, indicating the potential wide application of our method. We observed that the proposed method achieves greater improvement on heterophilic graphs compared to homophilic graphs, and heterophilic graphs are generally considered more challenging. This indicates that our method can effectively enhance the generalization capability of GNNs. We also provide additional experiments of integrating FGSAM+ with prompt-based FSNC [33] in the Appendix E.3.

## 5.7 Additional Study

We observe that both FGSAM and FGSAM+ generally outperform the standard SAM across tasks (FSNC and standard node classification). This is an interesting finding, as our FGSAM and FGSAM+ algorithm remove message-passing during the minimization of the perturbed loss, which is expected to hurt performance. We attribute these counter-intuitive results to the mitigation of the imbalance adversarial game. The training process of SAM-like

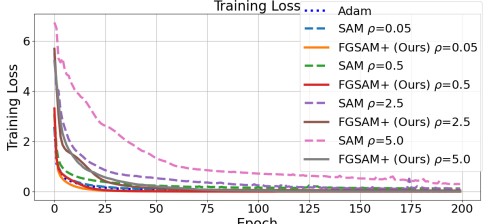

Figure 5: Training loss curves related to different $\rho$ across optimizers.

algorithms entails an adversarial game similar to that in Generative Adversarial Nets (GANs) [14]. Prior studies [3, 4, 28] have demonstrated that imbalanced adversarial games in GANs can give rise to worse results. Both FGSAM and FGSAM+ employ distinct models for perturbation and minimization, which can help alleviate the extent of imbalance. These factors may explain the observed performance discrepancies among the compared algorithms. To verify the explanation, we conduct experiments varying the hyper-parameter $\rho$. Specifically, we graphically illustrate the comparative training loss of SAM and FGSAM+ over a range of $\rho$ values in Fig. 5, which reveals that while SAM struggles to converge with higher $\rho$ values, FGSAM+ consistently achieves convergence. Moreover, it is established that a higher $\rho$ value is conducive to a tighter generalization bound, suggesting that a larger $\rho$ could potentially enhance performance. Consequently, FGSAM+ is capable of mitigating the imbalanced games issue and tolerating a larger $\rho$, which contributes to its enhanced performance.

## 6 Conclusion

In this work, we study the application of Sharpness-Aware Minimization (SAM) in FSNC to improve model's generalization, since the key for FSNC is to generalize the model to unseen samples. In order to alleviate the heavy computation cost of SAM, we utilize the connection between MLPs and GNNs and use MLPs to accelerate the training of GNNs. However, the low generalization and lack of using graph topology of MLPs also limit its performance. Hence we propose to apply GNNs to perturb parameters for generalization and use MLPs to minimize the perturbed training loss for conducting the proposed FGSAM. Moreover, we reuse the GNN gradient in perturbation in minimization for better including topology information. We further reduce the training time by conducting exact FGSAM update at every $k$ steps and approximate FGSAM's gradient with reusing information in the intermediate steps. Finally, the extensive experiments demonstrate the effectiveness and efficiency of our proposed methods.

## Acknowledgements

Jing Tang's work is partially supported by National Key R&D Program of China under Grant No. 2023YFF0725100, by the National Natural Science Foundation of China (NSFC) under Grant No. 62402410 and U22B2060, by National Language Commission under Grant No. WT145-39, by Guangdong Basic and Applied Basic Research Foundation under Grant No. 2023A1515110131, by Guangzhou Municipal Science and Technology Bureau under Grant No. 2023A03J0667 and 2024A04J4454, and by Createlink Technology Co., Ltd. Xiaochun Cao's work is supported in part by National Natural Science Foundation of China (No. 62411540034), in part by Shenzhen Science and Technology Program (Grant No. KQTD20221101093559018).

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

# A    Potential Broader Impact

This paper presents work whose goal is to advance the field of Machine Learning. There are many potential societal consequences of our work, none of which we feel must be specifically highlighted here.

# B    Algorithm

---

**Algorithm 1** Training with FGSAM and FGSAM+.

---

**Require:** $\mathcal{G}, \mathcal{C}_{\text{base}}$, learning rate $\eta$, radius $\rho$, FGSAM update interval $k$, adaptive ratio $\alpha$.
**Ensure:** A flat minimum solution $\hat{\boldsymbol{w}}$.
  Initialize weights $\boldsymbol{w}_0$;
  **for** $t \leftarrow 0$ **to** $T - 1$ **do**
    Sample training task $\mathcal{T}_t$ from $\mathcal{G}$ and $\mathcal{C}_{\text{base}}$;

    `### only for FGSAM`
    Vanilla grad $\boldsymbol{g}^{\text{gnn}} = \nabla_{\boldsymbol{w}_t} \mathcal{L}_{\mathcal{T}_t}(\boldsymbol{w}_t; f_{\text{gnn}})$;
    Perturbed weights $\hat{\boldsymbol{\epsilon}} = \rho \frac{\boldsymbol{g}^{\text{gnn}}}{\|\boldsymbol{g}^{\text{gnn}}\|}$;
    FGSAM-grad $\boldsymbol{g}_{\text{FGSAM}} = \lambda \boldsymbol{g}^{\text{gnn}} + \nabla_{\boldsymbol{w}_t} \mathcal{L}_{\mathcal{T}_t}(\boldsymbol{w}_t; f_{\text{mlp}})|_{\boldsymbol{w}_t + \hat{\boldsymbol{\epsilon}}}$;

    `### only for FGSAM+`
    **if** $t\%k = 0$ **then**
      `# the actual FGSAM-update`
      Vanilla grad $\boldsymbol{g}^{\text{gnn}} = \nabla_{\boldsymbol{w}_t} \mathcal{L}_{\mathcal{T}_t}(\boldsymbol{w}_t; f_{\text{gnn}})$;
      Vanilla grad $\boldsymbol{g}^{\text{mlp}} = \nabla_{\boldsymbol{w}_t} \mathcal{L}_{\mathcal{T}_t}(\boldsymbol{w}_t; f_{\text{mlp}})$;
      Perturbed weights $\hat{\boldsymbol{\epsilon}} = \rho \frac{\boldsymbol{g}^{\text{gnn}}}{\|\boldsymbol{g}^{\text{gnn}}\|}$;
      Topology-grad $\boldsymbol{g}_{\mathcal{G}} = \boldsymbol{g}^{\text{gnn}} - \|\boldsymbol{g}^{\text{gnn}}\| \frac{\boldsymbol{g}^{\text{gnn}} \cdot \boldsymbol{g}^{\text{mlp}}}{\|\boldsymbol{g}^{\text{gnn}}\|\|\boldsymbol{g}^{\text{mlp}}\|} \frac{\boldsymbol{g}^{\text{mlp}}}{\|\boldsymbol{g}^{\text{mlp}}\|}$;
      SAM-grad $\boldsymbol{g}_s = \nabla_{\boldsymbol{w}_t} \mathcal{L}_{\mathcal{T}_t}(\boldsymbol{w}_t; f_{\text{mlp}})|_{\boldsymbol{w}_t + \hat{\boldsymbol{\epsilon}}}$;
      Flatness-grad $\boldsymbol{g}_v = \boldsymbol{g}_s - \|\boldsymbol{g}_s\| \frac{\boldsymbol{g}^{\text{mlp}} \cdot \boldsymbol{g}_s}{\|\boldsymbol{g}^{\text{mlp}}\|\|\boldsymbol{g}_s\|} \frac{\boldsymbol{g}^{\text{mlp}}}{\|\boldsymbol{g}^{\text{mlp}}\|}$;
      FGSAM-grad $\boldsymbol{g}_{\text{FGSAM}} = \lambda \boldsymbol{g}^{\text{gnn}} + \boldsymbol{g}_s$;
    **else**
      `# approximate FGSAM-gradient`
      Vanilla grad $\boldsymbol{g}^{\text{mlp}} = \nabla_{\boldsymbol{w}_t} \mathcal{L}_{\mathcal{T}_t}(\boldsymbol{w}_t; f_{\text{mlp}})$;
      Approx gnn-grad $\hat{\boldsymbol{g}}^{\text{gnn}} = \boldsymbol{g}^{\text{mlp}} + \boldsymbol{g}_{\mathcal{G}} \frac{\|\boldsymbol{g}^{\text{mlp}}\|}{\|\boldsymbol{g}_{\mathcal{G}}\|}$
      Approx FGSAM-grad $\boldsymbol{g}_{\text{FGSAM}} = \boldsymbol{g}^{\text{mlp}} + \alpha \boldsymbol{g}_v \frac{\|\boldsymbol{g}^{\text{mlp}}\|}{\|\boldsymbol{g}_v\|} + \lambda \hat{\boldsymbol{g}}^{\text{gnn}}$;
    **end if**
    Update weights: $\boldsymbol{w}_{t+1} \leftarrow \boldsymbol{w}_t - \eta \cdot \boldsymbol{g}_{\text{FGSAM}}$;
  **end for**
  $\hat{\boldsymbol{w}} \leftarrow \boldsymbol{w}_T$.

---

# C    Proof

The linear classifier for K-classification problems can be formulated as $\frac{K(K-1)}{2}$ binary classification problems.

Hence we study the classification between class $C_o$ and $C_p$ without loss of generality.

The distribution of original features from different classes follows:

$$
\begin{aligned}
\boldsymbol{x}_i &\sim \mathcal{N}\left(\boldsymbol{\mu}_o, \boldsymbol{I}\right), i \in C_o \\
\boldsymbol{x}_i &\sim \mathcal{N}\left(\boldsymbol{\mu}_p, \boldsymbol{I}\right), i \in C_p
\end{aligned}
\tag{11}
$$

The distribution of filtered features from different classes follows:

$$h_i \sim \mathcal{N}\left(\frac{(p-q)\,\boldsymbol{\mu}_o + qK\bar{\boldsymbol{\mu}}}{p+(K-1)\,q}, \frac{\mathbf{I}}{deg\,(i)}\right), \quad i \in C_o,$$

$$h_i \sim \mathcal{N}\left(\frac{(p-q)\,\boldsymbol{\mu}_p + qK\bar{\boldsymbol{\mu}}}{p+(K-1)\,q}, \frac{\mathbf{I}}{deg\,(i)}\right), \quad i \in C_p,$$

(12)

For simplicity, we denote $\tilde{\boldsymbol{\mu}}_0 = \frac{(p-q)\boldsymbol{\mu}_o + qK\bar{\boldsymbol{\mu}}}{p+(K-1)q}$ and $\tilde{\boldsymbol{\mu}}_p = \frac{(p-q)\boldsymbol{\mu}_p + qK\bar{\boldsymbol{\mu}}}{p+(K-1)q}$.

Following [27], the optimal classifier of original features constructs a decision bound $\mathcal{P} = \{\boldsymbol{x}|\boldsymbol{w}^T\boldsymbol{x} - \boldsymbol{w}^T\boldsymbol{b}\}$, where $\boldsymbol{w} = \frac{\boldsymbol{\mu}_o - \boldsymbol{\mu}_p}{2}/||\frac{\boldsymbol{\mu}_o - \boldsymbol{\mu}_p}{2}||$, $\boldsymbol{b} = \frac{\boldsymbol{\mu}_o + \boldsymbol{\mu}_p}{2}$. Similarly, the optimal classifier of filtered features constructs a decision bound $\mathcal{P}' = \{\boldsymbol{h}|\boldsymbol{w}'^T\boldsymbol{h} - \boldsymbol{w}'^T\boldsymbol{b}\}$, where $\boldsymbol{w}' = \frac{\tilde{\boldsymbol{\mu}}_o - \tilde{\boldsymbol{\mu}}_p}{2}/||\frac{\tilde{\boldsymbol{\mu}}_o - \tilde{\boldsymbol{\mu}}_p}{2}||$, $\boldsymbol{b}' = \frac{\tilde{\boldsymbol{\mu}}_o + \tilde{\boldsymbol{\mu}}_p}{2}$.

And we have $\tilde{\boldsymbol{\mu}}_o - \tilde{\boldsymbol{\mu}}_p = \frac{p-q}{p+(K-1)q}(\boldsymbol{\mu}_o - \boldsymbol{\mu}_p)$, hence we have $\boldsymbol{w} = \boldsymbol{w}'$. Then we verify whether $\boldsymbol{w}^T\boldsymbol{b} = \boldsymbol{w}'^T\boldsymbol{b}'$:

$$
\begin{aligned}
\boldsymbol{w}'^T\boldsymbol{b}' &= \boldsymbol{w}'^T\left(\frac{\tilde{\boldsymbol{\mu}}_o + \tilde{\boldsymbol{\mu}}_p}{2}\right) \\
&= \boldsymbol{w}^T\frac{1}{2}\left(\frac{(p-q)\,\boldsymbol{\mu}_o + qK\bar{\boldsymbol{\mu}}}{p+(K-1)\,q} + \frac{(p-q)\,\boldsymbol{\mu}_p + qK\bar{\boldsymbol{\mu}}}{p+(K-1)\,q}\right) \\
&= \boldsymbol{w}^T\left(\frac{\lambda}{2}(\boldsymbol{\mu}_o + \boldsymbol{\mu}_p) + (1-\lambda)\bar{\boldsymbol{\mu}}\right) \\
&= \boldsymbol{w}^T\left(\frac{\lambda}{2}(\boldsymbol{\mu}_o + \boldsymbol{\mu}_p)\right. \\
&\qquad\left. + (1-\lambda)\frac{1}{2}(\bar{\boldsymbol{\mu}} - \boldsymbol{\mu}_o + \bar{\boldsymbol{\mu}} - \boldsymbol{\mu}_p + \boldsymbol{\mu}_o + \boldsymbol{\mu}_p)\right) \\
&= \boldsymbol{w}^T\left(\frac{1}{2}(\boldsymbol{\mu}_o + \boldsymbol{\mu}_p) + (1-\lambda)\left(\bar{\boldsymbol{\mu}} - \frac{\boldsymbol{\mu}_o + \boldsymbol{\mu}_p}{2}\right)\right) \\
&= \boldsymbol{w}^T\left(\frac{1}{2}(\boldsymbol{\mu}_o + \boldsymbol{\mu}_p)\right) + (1-\lambda)\boldsymbol{w}^T\left(\bar{\boldsymbol{\mu}} - \frac{\boldsymbol{\mu}_o + \boldsymbol{\mu}_p}{2}\right),
\end{aligned}
$$

(13)

where $\lambda = \frac{p-q}{p+(K-1)q}$.

Then we show $(\boldsymbol{\mu}_o - \boldsymbol{\mu}_p)^T\left(\bar{\boldsymbol{\mu}} - \frac{\boldsymbol{\mu}_o + \boldsymbol{\mu}_p}{2}\right) = \mathbf{0}$.

From $||\boldsymbol{\mu}_i - \boldsymbol{\mu}_j||_2 = D$, we have:

$$||\boldsymbol{\mu}_o - \bar{\boldsymbol{\mu}}||_2 = ||\boldsymbol{\mu}_p - \bar{\boldsymbol{\mu}}||_2,$$

(14)

which gives:

$$
\begin{aligned}
(\boldsymbol{\mu}_o - \bar{\boldsymbol{\mu}})^T(\boldsymbol{\mu}_o - \bar{\boldsymbol{\mu}}) &= (\boldsymbol{\mu}_p - \bar{\boldsymbol{\mu}})^T(\boldsymbol{\mu}_p - \bar{\boldsymbol{\mu}}) \\
\boldsymbol{\mu}_o^T\boldsymbol{\mu}_o - 2\boldsymbol{\mu}_o^T\bar{\boldsymbol{\mu}} + \bar{\boldsymbol{\mu}}^T\bar{\boldsymbol{\mu}} &= \boldsymbol{\mu}_p^T\boldsymbol{\mu}_o - 2\boldsymbol{\mu}_p^T\bar{\boldsymbol{\mu}} + \bar{\boldsymbol{\mu}}^T\bar{\boldsymbol{\mu}} \\
\boldsymbol{\mu}_o^T\boldsymbol{\mu}_o - 2\boldsymbol{\mu}_o^T\bar{\boldsymbol{\mu}} &= \boldsymbol{\mu}_p^T\boldsymbol{\mu}_o - 2\boldsymbol{\mu}_p^T\bar{\boldsymbol{\mu}}
\end{aligned}
$$

(15)

Hence we have:

$$
\begin{aligned}
&(\boldsymbol{\mu}_o - \boldsymbol{\mu}_p)^T\left(\bar{\boldsymbol{\mu}} - \frac{\boldsymbol{\mu}_o + \boldsymbol{\mu}_p}{2}\right) \\
&= \boldsymbol{\mu}_o^T\bar{\boldsymbol{\mu}} - \boldsymbol{\mu}_o^T\frac{\boldsymbol{\mu}_o + \boldsymbol{\mu}_p}{2} - \boldsymbol{\mu}_p^T\bar{\boldsymbol{\mu}} + \boldsymbol{\mu}_p^T\frac{\boldsymbol{\mu}_o + \boldsymbol{\mu}_p}{2} \\
&= \boldsymbol{\mu}_o^T\bar{\boldsymbol{\mu}} - \frac{1}{2}\boldsymbol{\mu}_o^T\boldsymbol{\mu}_o - \left(\boldsymbol{\mu}_p^T\bar{\boldsymbol{\mu}} - \frac{1}{2}\boldsymbol{\mu}_p^T\boldsymbol{\mu}_p\right) \\
&= \mathbf{0}
\end{aligned}
$$

(16)

Combining Eq. (13) and Eq. (16), we have:

$$\boldsymbol{w}'^T\boldsymbol{b}' = \boldsymbol{w}^T\boldsymbol{b},$$

(17)

which means $\mathcal{P} = \mathcal{P}'$.

This completes the proof.

Table 5: Statistics of evaluation datasets

| Datasets | # Nodes | # Edges | # Features | # Classes | Class Split |
|----------|---------|---------|-----------|-----------|-------------|
| **CoraFull** | 19,793 | 63,421 | 8,710 | 70 | 40/15/15 |
| **DBLP** | 40,672 | 288,270 | 7,202 | 137 | 80/27/30 |
| **ogbn-arXiv** | 169,343 | 1,157,799 | 128 | 40 | 20/10/10 |

Table 6: Hyper-parameters Search Space.

| Hyper-parameter | Search Space |
|-----------------|--------------|
| **MAML-based models:** | |
| learning rate | {0.05, 0.01, 0.001, 0.0001} |
| weight decay | {0.0, 0.001, 0.0005} |
| dropout | {0.0, 0.1, 0.3, 0.5, 0.7, 0.9} |
| $\rho$ | {0.01, 0.05, 0.1, 0.15, 0.2, 0.5, 0.8, 1.0, 1.2} |
| $\alpha$ | {0.5, 0.7, 0.9} |
| **non-MAML models:** | |
| learning rate finetune | {0.5, 0.1, 0.01, 0.001} |
| learning rate meta | {0.05, 0.01, 0.003, 0.001, 0.0001} |
| weight decay | {0.0, 0.001, 0.0005} |
| dropout | {0.0, 0.1, 0.3, 0.5, 0.7, 0.9} |
| $\rho$ | {0.01, 0.05, 0.1, 0.15, 0.2, 0.5, 0.8, 1.0, 1.2} |
| $\alpha$ | {0.5, 0.7, 0.9} |

# D   Experiments details

## D.1   Datasets Description

- **CoraFull** is an extension of the prevalent dataset 'Cora' [41], a citation network dataset. On this graph, nodes represent papers and edges represent citation links. The nodes are labeled on the paper topics. Node attributes are obtained using bag-of-words for the title and abstract of the paper.

- **DBLP** is also a citation network, where nodes represent papers and edges represent the citation between papers. Specifically, the node attributes are generated by the abstract and the node labels are based on the paper venues.

- **ogbn-arXiv** is a citation network among all Computer Science arXiv papers based on MAG [36]. Node represent papers and edges are citations links. The node attributes are obtained using skip-gram on abstract of papers. The nodes are labeled by the subject area.

## D.2   Implementation Details

Specifically, we implement our model by PyTorch [29] and conduct experiments on 24GB Nvidia RTX3090Ti, according to the training protocol Algorithm 2. Repeat number $R = 5$, patience $P = 10$, SAM update interval $k = 2$, validation interval $I = 10$, validation number $V = 20$, test number $W = 100$. For MAML models max epochs $T = 500$, and for non-MAML model max epochs $T = 1000$. We evaluate our method under various settings, i.e., $N = \{5, 10\}$, $K = \{3, 5\}$, but we set $N = \{2, 5\}$ for Coauthor-CS dataset. We use Optuna [2] for hyper-parameters searching for all models with various optimizers, the search space is shown in Tab. 6.

Note that we further split $\mathcal{C}_{\text{base}}$ into two disjoint class set: training class set $\mathcal{C}_{\text{tr}}$ and validation class set $\mathcal{C}_{\text{val}}$, such that $\mathcal{C}_{\text{base}} = \mathcal{C}_{\text{tr}} \cup \mathcal{C}_{\text{val}}$ and $\mathcal{C}_{\text{tr}} \cap \mathcal{C}_{\text{val}} = \emptyset$. Overall, we use $\mathcal{C}_{\text{tr}}$ and $\mathcal{C}_{\text{val}}$ for train and validation in the meta-training stage, respectively, and use $\mathcal{C}_{\text{novel}}$ for meta-test. We split $\mathcal{C}$ into $\mathcal{C}_{\text{tr}}$, $\mathcal{C}_{\text{val}}$ and $\mathcal{C}_{\text{novel}}$ according to the class split ratio in Tab. 5.

**Algorithm 2** Training Protocol of FSNC Task

---

**Require:** $\mathcal{G}$, $\mathcal{C}_{\mathrm{tr}}$, $\mathcal{C}_{\mathrm{val}}$, $\mathcal{C}_{\mathrm{novel}}$, repeat number $R$, max epochs $T$, patience $P$, validation interval $I$, validation number $V$, test number $W$.
**Ensure:** A trained model $\hat{f}$, model's accuracy $\hat{s}$.
  Initialize $f, s \leftarrow \{\}$.
  # repeat R times
  **for** $r \leftarrow 0$ to $R - 1$ **do**
    Initialize $s_{\mathrm{best}} \leftarrow 0$, $s_{\mathrm{test}} \leftarrow \{\}$, $p \leftarrow 0$;
    # meta-training
    **for** $t \leftarrow 0$ to $T - 1$ **do**
      # training
      Sample training task $\mathcal{T}_t = \{\mathcal{S}_t, \mathcal{Q}_t\}$ from $\mathcal{C}_{\mathrm{tr}}$;
      Optimize model $f$ on $\mathcal{T}_t$;
      # validation
      **if** $t\%I = 0$ **then**
        Sample $V$ validation tasks $\mathcal{T}_{\mathrm{val}}$ from $\mathcal{C}_{\mathrm{val}}$;
        Compute mean accuracy $s_{\mathrm{val}}$ on $\mathcal{T}_{\mathrm{val}}$ by $f$;
        **if** $s_{\mathrm{val}} > s_{\mathrm{best}}$ **then**
          $s_{\mathrm{best}} \leftarrow s_{\mathrm{val}}$, $p \leftarrow 0$;
        **else**
          $p \leftarrow p + 1$;
        **end if**
        # early-stop
        **if** $p = P$ **then**
          break;
        **end if**
      **end if**
    **end for**
    # meta-test
    Sample $W$ test tasks $\mathcal{T}_{\mathrm{test}}$ from $\mathcal{C}_{\mathrm{novel}}$;
    Compute mean accuracy $s_{\mathrm{test}}$ on these tasks using model $f$;
    $s = s \cup s_{\mathrm{test}}$;
  **end for**
  $\hat{f} \leftarrow f$, $\hat{s} \leftarrow \mathrm{mean}(s)$.

---

Table 7: Accuracy on the baseline with different optimizer. '5N3K' denotes 5-way 3-shot setting.

| | Corafull | | | | DBLP | | | | ogbn-arXiv | | | |
|---|---|---|---|---|---|---|---|---|---|---|---|---|
| | 5N3K | 5N5K | 10N3K | 10N5K | 5N3K | 5N5K | 10N3K | 10N5K | 5N3K | 5N5K | 10N3K | 10N5K |
| **Meta-GCN** | $70.25_{\pm2.09}$ | $77.00_{\pm2.36}$ | $51.19_{\pm2.86}$ | $58.85_{\pm2.61}$ | $82.60_{\pm1.32}$ | $85.20_{\pm3.41}$ | $65.96_{\pm4.16}$ | $70.85_{\pm1.89}$ | $49.32_{\pm3.26}$ | $54.37_{\pm6.27}$ | $30.68_{\pm3.06}$ | $28.20_{\pm10.09}$ |
| w/ SAM | $70.23_{\pm5.48}$ | $75.82_{\pm2.58}$ | $54.77_{\pm5.69}$ | $58.18_{\pm3.17}$ | $82.50_{\pm1.33}$ | $85.04_{\pm3.38}$ | $68.31_{\pm3.22}$ | $71.22_{\pm1.16}$ | $54.80_{\pm4.71}$ | $55.19_{\pm6.76}$ | $25.10_{\pm7.53}$ | $31.79_{\pm4.90}$ |
| w/ FGSAM | $70.97_{\pm3.15}$ | $77.64_{\pm2.00}$ | $55.53_{\pm4.35}$ | $59.30_{\pm2.96}$ | $82.66_{\pm1.34}$ | $85.26_{\pm3.36}$ | $69.22_{\pm2.87}$ | $71.80_{\pm1.91}$ | $52.45_{\pm3.33}$ | $57.05_{\pm4.67}$ | $28.92_{\pm10.19}$ | $31.03_{\pm5.04}$ |
| w/ FGSAM+ | $71.54_{\pm4.22}$ | $78.97_{\pm2.62}$ | $58.73_{\pm5.47}$ | $61.61_{\pm6.23}$ | $82.40_{\pm1.29}$ | $84.24_{\pm2.89}$ | $68.97_{\pm1.63}$ | $72.18_{\pm1.58}$ | $52.98_{\pm4.20}$ | $58.08_{\pm5.90}$ | $31.09_{\pm3.66}$ | $33.38_{\pm2.22}$ |
| **AMM-GNN** | $72.92_{\pm4.67}$ | $80.44_{\pm3.63}$ | $57.58_{\pm5.46}$ | $57.29_{\pm3.39}$ | $81.02_{\pm2.61}$ | $83.48_{\pm1.95}$ | $66.40_{\pm2.70}$ | $71.31_{\pm2.95}$ | $51.95_{\pm1.34}$ | $57.79_{\pm2.62}$ | $28.71_{\pm8.82}$ | $26.74_{\pm9.02}$ |
| w/ SAM | $68.47_{\pm3.02}$ | $74.10_{\pm2.82}$ | $52.43_{\pm2.78}$ | $57.94_{\pm3.69}$ | $80.54_{\pm2.50}$ | $83.45_{\pm2.03}$ | $66.29_{\pm2.71}$ | $71.50_{\pm3.02}$ | $49.42_{\pm5.06}$ | $50.75_{\pm6.84}$ | $30.57_{\pm6.25}$ | $32.42_{\pm4.42}$ |
| w/ FGSAM | $71.67_{\pm5.96}$ | $77.72_{\pm3.09}$ | $60.15_{\pm4.10}$ | $62.11_{\pm4.47}$ | $84.01_{\pm1.29}$ | $85.32_{\pm0.86}$ | $67.12_{\pm2.91}$ | $71.70_{\pm1.83}$ | $48.69_{\pm8.40}$ | $55.89_{\pm5.51}$ | $35.59_{\pm5.22}$ | $32.57_{\pm3.98}$ |
| w/ FGSAM+ | $72.79_{\pm4.44}$ | $79.18_{\pm2.19}$ | $59.59_{\pm6.05}$ | $62.61_{\pm3.99}$ | $81.24_{\pm1.66}$ | $85.07_{\pm2.26}$ | $70.37_{\pm4.86}$ | $71.32_{\pm0.84}$ | $51.02_{\pm6.38}$ | $50.49_{\pm9.12}$ | $33.60_{\pm3.32}$ | $34.05_{\pm3.47}$ |
| **GPN** | $65.23_{\pm1.30}$ | $65.67_{\pm3.40}$ | $50.48_{\pm3.24}$ | $51.23_{\pm5.72}$ | $76.05_{\pm1.19}$ | $75.02_{\pm3.53}$ | $65.41_{\pm3.03}$ | $64.52_{\pm3.22}$ | $55.35_{\pm5.01}$ | $57.50_{\pm4.72}$ | $42.72_{\pm5.10}$ | $41.54_{\pm7.95}$ |
| w/ SAM | $67.28_{\pm4.31}$ | $65.02_{\pm1.57}$ | $55.06_{\pm2.90}$ | $52.30_{\pm4.60}$ | $79.44_{\pm2.90}$ | $77.66_{\pm1.76}$ | $67.88_{\pm1.28}$ | $67.78_{\pm2.59}$ | $56.18_{\pm1.86}$ | $58.65_{\pm4.34}$ | $39.91_{\pm6.81}$ | $39.92_{\pm2.99}$ |
| w/ FGSAM | $69.54_{\pm3.11}$ | $69.37_{\pm1.07}$ | $57.85_{\pm5.03}$ | $56.49_{\pm4.42}$ | $80.10_{\pm2.69}$ | $79.61_{\pm2.27}$ | $68.50_{\pm2.22}$ | $69.44_{\pm1.78}$ | $57.58_{\pm4.56}$ | $58.23_{\pm3.95}$ | $47.67_{\pm3.97}$ | $48.20_{\pm3.73}$ |
| w/ FGSAM+ | $69.40_{\pm4.57}$ | $69.96_{\pm2.95}$ | $57.74_{\pm4.17}$ | $56.10_{\pm3.36}$ | $80.02_{\pm1.89}$ | $79.69_{\pm2.24}$ | $68.94_{\pm1.99}$ | $69.51_{\pm2.54}$ | $57.39_{\pm3.36}$ | $58.04_{\pm2.40}$ | $46.59_{\pm3.24}$ | $49.49_{\pm3.74}$ |
| **TENT** | $71.24_{\pm2.05}$ | $75.49_{\pm1.88}$ | $57.29_{\pm4.00}$ | $60.35_{\pm2.80}$ | $80.67_{\pm3.19}$ | $82.74_{\pm1.84}$ | $69.04_{\pm2.45}$ | $71.79_{\pm2.68}$ | $60.44_{\pm5.48}$ | $67.34_{\pm2.15}$ | $47.14_{\pm4.25}$ | $54.88_{\pm4.97}$ |
| w/ SAM | $71.38_{\pm2.47}$ | $75.29_{\pm4.09}$ | $56.86_{\pm2.28}$ | $61.85_{\pm2.89}$ | $82.13_{\pm2.02}$ | $85.10_{\pm0.54}$ | $68.96_{\pm3.85}$ | $73.62_{\pm1.56}$ | $63.58_{\pm2.18}$ | $69.30_{\pm3.48}$ | $50.79_{\pm3.15}$ | $55.21_{\pm2.57}$ |
| w/ FGSAM | $71.10_{\pm4.79}$ | $76.72_{\pm3.00}$ | $57.86_{\pm3.26}$ | $63.71_{\pm4.32}$ | $82.99_{\pm2.25}$ | $86.13_{\pm0.52}$ | $70.31_{\pm1.92}$ | $73.41_{\pm1.45}$ | $63.88_{\pm1.64}$ | $71.15_{\pm2.43}$ | $53.32_{\pm1.94}$ | $57.08_{\pm3.52}$ |
| w/ FGSAM+ | $72.85_{\pm4.14}$ | $77.77_{\pm3.44}$ | $58.37_{\pm4.13}$ | $63.04_{\pm3.56}$ | $83.64_{\pm1.55}$ | $85.97_{\pm0.56}$ | $71.15_{\pm2.43}$ | $73.72_{\pm1.05}$ | $66.20_{\pm4.41}$ | $69.14_{\pm1.96}$ | $50.66_{\pm1.68}$ | $53.56_{\pm1.93}$ |

### D.3 Evaluation Results with Standard Deviation

In Tab. 7 and Tab. 8 we present the detailed results of Tab. 2 and Tab. 7 with standard deviation, respectively.

Table 8: Results on nine real-world node classification benchmark datasets: Mean accuracy (%). The best results are denoted as **boldface**.

| Model | Optimizer | Cora | Citeseer | Pubmed | Chameleon | Squirrel | Actor | Cornell | Texas | Wisconsin |
|---|---|---|---|---|---|---|---|---|---|---|
| GCN | Adam | $88.36_{\pm1.50}$ | $77.25_{\pm0.80}$ | $88.71_{\pm0.45}$ | $65.04_{\pm2.87}$ | $52.49_{\pm2.30}$ | $28.54_{\pm0.88}$ | $61.08_{\pm5.57}$ | $60.27_{\pm3.51}$ | $55.29_{\pm2.75}$ |
|  | SAM | $88.42_{\pm1.35}$ | $77.30_{\pm0.85}$ | $88.79_{\pm0.45}$ | $65.57_{\pm2.29}$ | $52.51_{\pm2.07}$ | $28.59_{\pm0.75}$ | $61.89_{\pm2.02}$ | $62.70_{\pm4.99}$ | $54.51_{\pm4.42}$ |
|  | **FGSAM** | $88.36_{\pm1.51}$ | $77.60_{\pm0.69}$ | $89.36_{\pm0.49}$ | $66.16_{\pm2.96}$ | $53.95_{\pm1.48}$ | $29.88_{\pm1.06}$ | $67.30_{\pm3.83}$ | $63.24_{\pm5.10}$ | $55.69_{\pm4.31}$ |
|  | **FGSAM+** | $88.32_{\pm1.48}$ | $77.52_{\pm0.96}$ | $89.13_{\pm0.44}$ | $64.56_{\pm2.56}$ | $51.14_{\pm2.36}$ | $29.66_{\pm0.78}$ | $68.11_{\pm4.37}$ | $61.62_{\pm6.22}$ | $54.71_{\pm5.48}$ |
| GraphSAGE | Adam | $87.67_{\pm1.96}$ | $76.09_{\pm1.43}$ | $89.15_{\pm0.57}$ | $50.33_{\pm1.97}$ | $37.61_{\pm1.18}$ | $33.74_{\pm1.16}$ | $78.11_{\pm5.95}$ | $78.38_{\pm5.47}$ | $84.51_{\pm3.51}$ |
|  | SAM | $87.69_{\pm1.71}$ | $76.44_{\pm1.21}$ | $89.25_{\pm0.51}$ | $50.92_{\pm2.26}$ | $37.44_{\pm1.08}$ | $33.83_{\pm1.09}$ | $78.92_{\pm4.40}$ | $80.27_{\pm4.71}$ | $84.31_{\pm4.42}$ |
|  | **FGSAM** | $88.36_{\pm1.51}$ | $77.13_{\pm0.69}$ | $89.75_{\pm0.49}$ | $51.34_{\pm2.96}$ | $39.12_{\pm1.48}$ | $34.53_{\pm1.06}$ | $82.43_{\pm3.83}$ | $81.35_{\pm5.10}$ | $86.47_{\pm4.31}$ |
|  | **FGSAM+** | $88.16_{\pm1.85}$ | $77.21_{\pm1.26}$ | $89.71_{\pm0.39}$ | $50.94_{\pm1.94}$ | $38.87_{\pm1.81}$ | $34.70_{\pm0.82}$ | $81.35_{\pm5.54}$ | $79.46_{\pm4.65}$ | $86.47_{\pm4.34}$ |
| GAT | Adam | $88.32_{\pm1.59}$ | $76.37_{\pm0.90}$ | $87.48_{\pm0.37}$ | $46.51_{\pm2.96}$ | $31.46_{\pm1.01}$ | $29.45_{\pm0.88}$ | $59.19_{\pm3.63}$ | $62.16_{\pm4.43}$ | $55.49_{\pm5.49}$ |
|  | SAM | $88.49_{\pm1.74}$ | $76.78_{\pm0.84}$ | $87.24_{\pm0.53}$ | $46.82_{\pm2.80}$ | $31.61_{\pm1.35}$ | $29.49_{\pm0.78}$ | $59.46_{\pm4.02}$ | $62.16_{\pm4.26}$ | $55.29_{\pm6.93}$ |
|  | **FGSAM** | $88.60_{\pm1.51}$ | $76.98_{\pm0.69}$ | $87.63_{\pm0.49}$ | $47.87_{\pm2.96}$ | $32.35_{\pm1.48}$ | $30.41_{\pm1.06}$ | $61.89_{\pm3.83}$ | $65.95_{\pm5.10}$ | $59.41_{\pm4.31}$ |
|  | **FGSAM+** | $88.70_{\pm1.82}$ | $77.10_{\pm1.12}$ | $87.74_{\pm0.56}$ | $48.07_{\pm3.25}$ | $32.69_{\pm2.34}$ | $30.60_{\pm0.99}$ | $62.16_{\pm2.91}$ | $64.86_{\pm5.95}$ | $58.04_{\pm5.05}$ |

Table 9: Time consumption comparison. The results stands for the time (sec.) consumed in 200 episodes training (only consider the feed-forward and -backward).

| Setting | Corafull | | | | DBLP | | | | ogbn-arXiv | | | |
|---|---|---|---|---|---|---|---|---|---|---|---|---|
|  | 5N3K | 5N5K | 10N3K | 10N5K | 5N3K | 5N5K | 10N3K | 10N5K | 5N3K | 5N5K | 10N3K | 10N5K |
| **Meta-GCN** | 9.56 | 9.58 | 9.38 | 9.40 | 17.61 | 17.60 | 17.50 | 17.59 | 41.09 | 40.98 | 40.96 | 40.92 |
| w/ SAM | 19.12 | 19.16 | 18.84 | 18.99 | 35.38 | 35.38 | 35.17 | 35.28 | 82.54 | 82.65 | 82.40 | 82.58 |
| w/ **FGSAM** | 10.91 | 10.82 | 10.79 | 10.80 | 19.15 | 19.22 | 19.11 | 19.12 | 42.50 | 42.54 | 42.45 | 42.45 |
| w/ **FGSAM+** | 6.58 | 6.48 | 6.51 | 6.48 | 10.77 | 10.77 | 10.51 | 10.44 | 22.17 | 22.21 | 22.02 | 22.06 |
| **AMM-GNN** | 15.03 | 15.04 | 14.94 | 15.00 | 26.71 | 26.74 | 26.72 | 26.76 | 42.27 | 42.39 | 42.30 | 42.37 |
| w/ SAM | 30.91 | 30.93 | 30.66 | 30.83 | 54.63 | 54.85 | 54.69 | 54.87 | 84.71 | 85.12 | 84.77 | 85.13 |
| w/ **FGSAM** | 17.55 | 17.89 | 17.51 | 17.44 | 30.08 | 30.13 | 30.28 | 30.16 | 44.32 | 44.45 | 44.43 | 44.46 |
| w/ **FGSAM+** | 9.99 | 10.05 | 9.97 | 9.99 | 16.14 | 16.44 | 16.25 | 16.22 | 23.24 | 23.24 | 23.13 | 23.15 |
| **GPN** | 1.84 | 1.93 | 1.87 | 1.92 | 3.26 | 3.26 | 3.29 | 3.29 | 7.67 | 7.74 | 7.67 | 7.73 |
| w/ SAM | 3.68 | 3.69 | 3.55 | 3.58 | 6.76 | 6.80 | 6.76 | 6.81 | 15.90 | 16.02 | 15.95 | 16.06 |
| w/ **FGSAM** | 2.33 | 2.19 | 2.40 | 2.39 | 3.97 | 3.95 | 4.03 | 4.04 | 8.56 | 8.58 | 8.58 | 8.58 |
| w/ **FGSAM+** | 1.62 | 1.48 | 2.06 | 2.14 | 2.43 | 2.51 | 2.59 | 2.70 | 4.68 | 4.66 | 4.64 | 4.65 |
| **TENT** | 7.58 | 8.29 | 13.14 | 14.49 | 7.79 | 8.70 | 13.65 | 15.30 | 9.21 | 9.98 | 15.53 | 16.87 |
| w/ SAM | 15.05 | 16.89 | 26.67 | 29.50 | 15.98 | 17.49 | 27.70 | 30.25 | 19.28 | 20.53 | 31.78 | 34.12 |
| w/ **FGSAM** | 14.85 | 14.63 | 25.97 | 25.68 | 15.28 | 15.40 | 26.52 | 26.59 | 17.57 | 17.60 | 29.08 | 29.34 |
| w/ **FGSAM+** | 11.13 | 11.04 | 19.10 | 19.15 | 11.47 | 11.50 | 19.69 | 19.64 | 12.53 | 12.61 | 21.14 | 21.15 |

## D.4 The Full Results of Time Consumption

Here we present the detailed results of training time consumption of different optimizers across various datasets. Tab. 9 indicates that our proposed algorithm FGSAM demonstrates only a slight increase in training cost compared to Adam in most cases. Furthermore, our enhanced version FGSAM+ outperforms Adam in terms of speed in the majority of scenarios. It is worth mentioning that our proposed algorithms achieve superior or comparable performance when compared to both Adam and SAM.

As mentioned before, for models composed of many non-GNN components (e.g. TENT), the training time on FGSAM+ may be still longer than that on Adam, since it is hardly further reduced.

# E Additional Experiments

## E.1 Statistics Of Benchmark Datasets In Node Classification

Table 10: Benchmark datasets statistics for node classification

|  | Cora | Citeseer | Pubmed | Chameleon | Squirrel | Actor | Cornell | Texas | Wisconsin |
|---|---|---|---|---|---|---|---|---|---|
| # Nodes | 2708 | 3327 | 19717 | 2277 | 5201 | 7600 | 183 | 183 | 251 |
| # Edges | 5278 | 4552 | 44324 | 18050 | 108536 | 15009 | 149 | 162 | 257 |
| # Classes | 7 | 6 | 3 | 5 | 5 | 5 | 5 | 5 | 5 |
| # Features | 1433 | 3703 | 500 | 2325 | 2089 | 932 | 1703 | 1703 | 1703 |
| $\mathcal{H}(\mathcal{G})$ | 0.81 | 0.74 | 0.80 | 0.28 | 0.24 | 0.38 | 0.57 | 0.41 | 0.45 |

Table 11: Performance of different update interval $k$.

| | | Corafull | | DBLP | |
|---|---|---|---|---|---|
| | | acc (%) | time (s) | acc (%) | time (s) |
| GPN w/ FGSAM | | 69.54 | 2.33 | 80.10 | 3.97 |
| GPN w/ FGSAM+ | 2 | 69.40 | 1.62 | 80.02 | 2.43 |
| | 5 | 70.02 | 0.93 | 78.10 | 1.49 |
| | 10 | 67.03 | 0.69 | 75.91 | 1.24 |

Table 12: Performance of prompt-based FSNC on Citeseer.

| Setting | 3 shots | | 5 shots | |
|---|---|---|---|---|
| | acc (%) | F1 | acc (%) | F1 |
| ProG [33] | 59.50 | 57.75 | 76.50 | 76.61 |
| **FGSAM+** | 60.33 | 58.43 | 77.00 | 77.21 |

## E.2 The Effect of Update Interval $k$ in FGSAM+

Here we study the effect of update interval $k$ in FGSAM+. It can be observed from Tab. 11 that as $k$ increases, the performance decreases, but meanwhile training time also decreases. This indicates that the possibility of choosing k to achieve a better trade-off between performance and efficiency. We note that the performance drop with increasing k seems to be larger compared to LookSAM [23] in computer vision tasks. This indicates the importance of the perturbation step in FGSAM+, as it not only introduces information about flat minima, but also incorporates neighbor information in training. Therefore, we recommend setting $k = 2$ as the prior optimal update interval to avoid large information loss.

## E.3 Integrating with Prompt-Based FSNC

Recently, there are many prompt-based methods [25, 33] have been developed, showing promising performance in FSNC. Hence, we investigate how our method performs in such a prompt-based FSNC task. Note that under this setting, the proposed method is used in prompt tuning instead of training. As shown in Tab. 12, our method improves the baseline [33] with a remarkable margin.

