# OpenReview forum: "Fast Graph Sharpness-Aware Minimization for Enhancing and Accelerating Few-Shot Node Classification"
_NeurIPS.cc/2024/Conference — NeurIPS 2024 poster_

### Official Review · Reviewer_phcW · 2024-07-14

**Soundness:** 2
**Presentation:** 3
**Contribution:** 2
**Rating:** 5
**Confidence:** 3

**Summary:**

The paper introduces an innovative approach to improve the generalization capabilities of Graph Neural Networks (GNNs) in few-shot node classification tasks. The authors propose a novel algorithm, Fast Graph Sharpness-Aware Minimization (FGSAM), which incorporates sharpness-aware minimization (SAM) techniques into GNN training but reduces the typical computational overhead by integrating multilayer perceptrons (MLPs) for efficiency. This method not only demonstrates superior performance on few-shot learning tasks compared to traditional techniques but also offers significant computational advantages, particularly in reducing training times.

**Strengths:**

originality: medium
quality: medium
clarity: medium
significance: medium

**Weaknesses:**

Some experiments are missing and the proof need to be double checked carefully.

**Questions:**

1. What is the standard deviations for the results in table 4? Are the improvements significant?

2. Is there any theoretical justification for PeerMLP?

3. What is g in equation (8)?

4. The proof of theorem 4.1 is confusing. Do you assume all nodes have the same degree deg(i)?

**Limitations:**

the authors adequately addressed the limitations

---

> ### Author Rebuttal · Authors · 2024-08-07
>
> Thank you for your valuable comments and reviews. We provide a point-by-point response below. Hope this can address your concern and make things clearer.
>
> > **Q1:** What is the standard deviations for the results in table 4? Are the improvements significant?
>
> **Response to Q1:** The detailed standard deviations for the results presented in Table 4 are provided in Table 8, located in Appendix D.3. In order to assess the significance of the observed improvements, we performed paired t-tests on the results obtained from ten repeated trials comparing SAM with FGSAM and FGSAM+, respectively, across each dataset. The null hypothesis for these tests was that the performance of SAM would not be inferior to that of FGSAM or FGSAM+. The outcomes of these statistical tests are encapsulated in the **Table A** and **Table B**. In these tables, the plus symbol denotes levels of statistical significance, with +, ++, and +++ corresponding to significance levels of 5\%, 1\%, and 0.1\%, respectively. The results indicate that both FGSAM and FGSAM+  achieve statistically significant improvement compared with SAM in most cases.
>
> **Table A. FGSAM vs SAM**
> |          | Cora | Citeseer | Pubmed | Chameleon | Squirrel | Actor | Cornell | Texas | Wisconsin |
> |-----------|------|----------|--------|-----------|----------|-------|---------|-------|-----------|
> | **GCN**       | -    | +        | +++    | +         | ++       | +++   | +++     | ++    | ++        |
> | **GraphSAGE** | +++  | +++      | +++    | +++       | +++      | +++   | +++     | +++   | +++       |
> | **GAT**       | +    | ++       | ++     | ++        | +++      | +++   | +++     | +++   | +++       |
>
> **Table B. FGSAM+ vs SAM**
> |          | Cora | Citeseer | Pubmed | Chameleon | Squirrel | Actor | Cornell | Texas | Wisconsin |
> |-----------|------|----------|--------|-----------|----------|-------|---------|-------|-----------|
> | **GCN**       | -    | +        | +++    | -         | ++       | ++    | +++     | -     | +         |
> | **GraphSAGE** | ++   | +++      | +++    | -         | +++      | +++   | +++     | -     | +++       |
> | **GAT**       | +    | ++       | +++    | ++        | +++      | +++   | +++     | +++   | +++       |
>
>
> > **Q2:** Is there any theoretical justification for PeerMLP?
>
> **Response to Q2:** Our Theorem 4.1 in the paper demonstrates that under the linear case, whether the MP  layer is used or not, the optimal decision bound is the same in the context of K-classes CSBM. We believe this theorem can give some valuable insight in using PeerMLP. Moreover, our approach reintroduces the graph topology in minimization. This is essentially different from PeerMLP which does not involve any graph topology information in training.
>
>
> > **Q3:** What is g in equation (8)?
>
> **Response to Q3:** The term $g$ is defined as $\nabla_w L_{D_{\text{tr}}}(w)$, referred to line 189 in page5. This represents the *vanilla gradient* of the loss function $L$ with respect to the weights $w$, computed on the training dataset $D_{\text{tr}}$.
>
>
> > Q4: The proof of theorem 4.1 is confusing. Do you assume all nodes have the same degree deg(I)?
>
> **Response to Q4:** We do not assume all nodes having the same degree. The theorem and its proof are based on the definition of the K-classes CSBM, as detailed between Lines 218 and 226 of our manuscript. In the context of CSBM, while it is true that the expected degree of nodes within the same class is identical, it is crucial to note that this does not necessitate uniformity in the actual degrees across all nodes.

---

> > ### Comment · Reviewer_phcW · 2024-08-13
> >
> > Thanks for the response. The authors addressed  some of my concerns and I will increase  my score to 5

---

> ### Author Response · Authors · 2024-08-12
>
> Dear Reviewer phcW,
>
> We hope our rebuttal sufficiently addressed your concerns. Is there any additional information we can provide that might lead you to increase your rating? We look forward to your feedback.
>
> Many thanks,
>
> Author

---

> ### Author Response · Authors · 2024-08-14
>
> We thank you for acknowledging our work and for raising the score. Thanks again for your time and effort in reviewing our work.

---

### Official Review · Reviewer_4EHj · 2024-07-17

**Soundness:** 3
**Presentation:** 3
**Contribution:** 2
**Rating:** 5
**Confidence:** 3

**Summary:**

This paper focuses on efficient graph neural network (GNN) training in few-shot node classification (FSNC) problem by extending sharpness-aware minimization (SAM) for reducing the computational cost and improving the generalization of GNNs on unseen classes. The training phase is accelerated by perturbing the parameters of GNN and then minimizing the perturbation loss of GNN without the message passing mechanism (MLP). Experiments have been conducted to verify the effectiveness and efficiency of the proposed method.

**Strengths:**

1. It is interesting to incorporate SAM technique with GNN by removing MP during training and reintroducing MP in inference, which is reasonable to improve the generalization of FSNC.
2. The proposed method could be conveniently integrated into existing GNN-based few-shot node classification models, which is verified in the experiments.
3. The carefully designed experiments highlight the effectiveness of the proposed method in reducing the computational costs of GNN training.
4. The paper is well-organized and explains the method very clearly. In addition, the landscape visualization and toy case analyses also make the motivation and ideas easy to understand.

**Weaknesses:**

1. There are related work on applying SAM to few-shot tasks (Sharp-MAML) [1] and performing SAM every $k$ steps [2], and the innovation of this work is not significant compared with the above work.

[1] Sharp-maml: Sharpness-aware model-agnostic meta learning. ICML, 2022

[2] Towards efficient and scalable sharpness-aware minimization. CVPR, 2022

2. The experiments only compare with small-sample node classification models and variants of SAM from 2022, lacking comparisons with the latest baseline methods of FSNC.

**Questions:**

1. Since this work focuses on solving FSNC problems, the authors should provide a comparison with the state-of-the-art FSNC methods, such as COSMIC [3] and COLA [4], in theoretical analysis and experiments.

[3] Contrastive meta-learning for few-shot node classification. SIGKDD, 2023

[4] Graph contrastive learning meets graph meta learning: A unified method for few-shot node tasks. WWW, 2024

2. Why adding 25% noisy edges yields better results than adding only 15%? in Figure 4? The authors should discuss or explain this issue in details.

**Limitations:**

The authors describe some limitations of the proposed method in time consumption section.

---

> ### Author Rebuttal · Authors · 2024-08-07
>
> Thank you for your valuable comments and reviews. We provide a point-by-point response below. Hope this can address your concern and make things clearer.
>
> > **W1:** There are related work on applying SAM to few-shot tasks (Sharp-MAML) [1] and performing SAM every k steps [2], and the innovation of this work is not significant compared with the above work.
>
> **Response to W1:**
> Both Sharp-MAML [1] and LookSAM [2] focus on vision data, while our work is crafted in the context of graph data. Specifically, we utilize GNNs for parameter perturbation while employing MLPs to minimize the perturbed loss.
> Our experimental results demonstrate that directly applying SAM-like algorithms from the vision domain to graphs does not yield satisfactory performance (refer to Table 2, Table 3, and Figure 5), while our method is not only faster but also better. This indicates the necessity for SAM variants that are specifically designed for graph data.
> Furthermore, although sharp-MAML applies SAM to few-shot tasks, their algorithm is only designed for MAML models. In contrast, our work can be applied to both MAML and non-MAML methods in GNN-based FSNC tasks.
> Most notably, from the perspective of SAM, our work is crafted for graphs by its unique property, enabling the first SAM-like algorithm that can be faster than the base optimizer. This effectively turns SAM's core drawback of slower training speed into an advantage, distinguishing our work from previous SAM-like works.
>
> > **Q1:** Lacking comparisons with the latest baseline methods of FSNC such as COSMIC [3] and COLA [4], in theoretical analysis and experiments.
>
> **Response to Q1:**
> Our proposed FGSAM is fundamentally orthogonal to GNN-based FSNC models, such as COSMIC and COLA, due to its unique positioning as a SAM-like optimizer designed to leverage the intrinsic properties of GNNs.
> We selected classical and widely used FSNC models and NC models for evaluation. The strategy is borrowed from previous SAM works, where they also select classical and widely used models in the vision domain. We believe the selected baseline is diverse enough to verify the effectiveness of our proposed methods.
>
> However, we are glad to provide additional experiments on integrating FGSAM with state-of-the-art FSNC methods, such as COSMIC [3].
>
> |           | Corafull |  | DBLP  |  |
> |-----------|----------|------|-------|------|
> |           | acc      | time | acc   | time |
> | **COSMIC**   | 75.74    | 3.95 | 80.80 | 3.78 |
> | **COSMIC+FGSAM**  | 77.11    | 5.66 | 81.93 | 5.00 |
> | **COSMIC+FGSAM+** | 76.99    | 3.95 | 81.46 | 3.35 |
>
> Due to the time limitations, we evaluate our proposed method on COSMIC with 5N3K setting only on CoraFull and DBLP. As shown in the above table, FGSAM/FGSAM+ can effectively improve the performance of COSMIC, demonstrating the superiority of our proposed method.
>
> > **Q2:** Why adding 25\% noisy edges yields better results than adding only 15\%? in Figure 4?
>
> **Response to Q2:**
> Firstly, we found a typo in the coordinate axis labels in the original manuscript. The correct coordinate axis should be 0\%, 1\%, 5\%, 10\%, 15\%, not as previously misstated. Consequently, the observed phenomenon is that adding 10\% noisy edges yields better results than adding only 5\%.
>
> We appreciate the reviewer's interest in this intriguing phenomenon. In response, we conduct similar experiments on the AMM-GNN to validate the consistency of this phenomenon across different models. The results are presented in the following table:
>
> | edge noise         | 0\%   | 1\%   | 5\%   | 10\%  | 15\%  |
> |--------------------|-------|-------|-------|-------|-------|
> | **AMM-GNN**   | 72.92 | 70.50 | 69.98 | 70.33 | 68.41 |
> | **AMM-GNN+FGSAM+** | 72.79 | 71.94 | 70.40 | 70.85 | 68.73 |
>
>
> It can be seen that adding 10\% noisy edge also yields better results than adding 5\%, similar to the results on GPN. Hence, we suppose that this phenomenon may be attributed to the uniform introduction of a 10\% noisy edge, which perhaps aligns more closely with certain inherent characteristics of the dataset, thereby mitigating the extent of performance degradation.

---

> > ### Comment · Reviewer_4EHj · 2024-08-12
> >
> > Thanks for your feedback, which has addressed my concerns on the comparison methods and the results of adding noisy edges. Thus, I will maintain my score.

---

> ### Author Response · Authors · 2024-08-14
>
> We thank you for acknowledging our work. Thanks again for your time and effort in reviewing our work.

---

### Official Review · Reviewer_nij7 · 2024-07-21

**Soundness:** 3
**Presentation:** 3
**Contribution:** 3
**Rating:** 8
**Confidence:** 4

**Summary:**

The paper proposes a method for few-shot learning on graphs leveraging sharpness-aware minimization (SAM) from the vision community. The paper explains SAM as a technique for gradient perturbation during training to push the parameters to "flatter" regions of the loss space in hopes of achieving better generalization in the few shot setting. However, SAM is inherintely slower, requiring two backward/forward passes during training; one to compute the loss gradient and a second to determine the perturbation direction. The authors propose multiple ideas to overcome the added computation and to leverage graph topology. The authors strategically substitute Graph Neural Networks (GNNs) with Multilayer Perceptions (MLPs) to avoid the burden of message passing in some parts of their algorithm while preserving graph topologies in others. The authors propose a second scheme that leverages iterations of an approximate perturbation scheme with periodic exact evaluations of SAM. The authors present ablation studies on multiple SOTA models and 9 datasets and show improvements in training times and accuracy.

**Strengths:**

The authors present multiple novel approaches to overcome computational challenges applying SAM techniques to structured data to achieve better generality in the few shot case. I only have a few style suggestions at the beginning of the paper but otherwise the paper does a good job of explaining multiple, complex ideas in a clear way. The paper does present a significant result, blending ideas from PeerML and SAM from the vision community to contribute to making structured models more generalizable.

**Weaknesses:**

I think the paper is very good and clear in most cases. The only weaknesses I would note (and this may be subjective) would be to make some style improvements in the beginning of the paper.

(1) In the abstract (line 14) Moreover, our method ingeniously reutilizes... is a bit grandiose; I would prefer a more scientific tone and delete "Moreover" and "ingeniously"; "Our method reutilizies...

(2) Remove "for the first time" in 20. It is clear that the ideas are novel.

(3) Line 50 is the first time Message Passing is abbreviated as MP. I would follow the convention used in the paper for the introduction of other abbreviations where the first letter of the abbreviated character is in bold and capitalized.

(4) Line 93 "as follows" is not followed by an equation but another sentence. Maybe the equation should be presented after the "as follows" text and the next sentence come after the equation?

**Questions:**

(1) I find the paragraph containing lines 137-145 hard to read due to how the ideas are presented:

"Inspired by previous work...we propose to remove MP during training, but reintroduce it..Reintroducing MP after training can improve the performance significanly but still cannot surpass GNNs...""Henve we propose minimizing training loss on PeerMLPs but minimizing the sharpness according to GNNs.

There seems to be some logical oscillation in the text. First it seems that a scheme similar to PeerMLP is uses where the GNN is removed during training and injected back during inference. In the next few sentences, this seems to be abandoned and the GNN brought back for the SAM portion of training. I believe the experiments and ideas really did implement the latter formulation but either I'm misunderstanding or this paragraph could use some work to be clearer.

(2) In table 8 in the Appendix, I don't believe the best results are actually in boldface. Maybe this was due to my printer or perhaps the table needs to be reformatted?

**Limitations:**

No limitations.

---

> ### Author Rebuttal · Authors · 2024-08-07
>
> We sincerely appreciate your constructive feedback and valuable comments on our manuscript. We found that your comments are really valuable to further improve our manuscript. Below, we address each of your concerns and questions:
>
> > **Q1:** I think the paper is very good and clear in most cases. The only weaknesses I would note (and this may be subjective) would be to make some style improvements in the beginning of the paper.
>
> **Response to Q1:** Thanks for your careful pointing out. We will improve our writing in the revision.
>
> > **Q2:** I find the paragraph containing lines 137-145 hard to read due to how the ideas are presented:
> "Inspired by previous work...we propose to remove MP during training, but reintroduce it..Reintroducing MP after training can improve the performance significantly but still cannot surpass GNNs...""Henve we propose minimizing training loss on PeerMLPs but minimizing the sharpness according to GNNs.
> There seems to be some logical oscillation in the text. First it seems that a scheme similar to PeerMLP is uses where the GNN is removed during training and injected back during inference. In the next few sentences, this seems to be abandoned and the GNN brought back for the SAM portion of training. I believe the experiments and ideas really did implement the latter formulation but either I'm misunderstanding or this paragraph could use some work to be clearer.
>
> **Response to Q2:** Thank you for your valuable feedback, which has highlighted an area of our text that could benefit from greater clarity. In response, we would like to clarify that our approach indeed adopts the latter formulation mentioned, wherein MP is partially removed during training, with the reintroduction of GNN for the SAM portion of the training. The intent of our original discussion was to illustrate the relative performance of PeerMLP, acknowledging its merits yet also its limitations, thereby motivating our proposed method. To address the issue raised, we will revise the mentioned text to ensure the presentation of our idea is more clear and logical.
>
> > **Q3:** In Table 8 in the Appendix, I don't believe the best results are actually in boldface. Maybe this was due to my printer or perhaps the table needs to be reformatted?
>
> **Response to Q3:** Thanks for pointing out this! Upon review, we found that the boldfacing issue in Table 8 was indeed a formatting oversight on our part. We will correct this in the revised manuscript.

---

> > ### Comment · Reviewer_nij7 · 2024-08-12
> > **Addressed concerns**
> >
> > Thank you for addressing the concerns. Best of luck with the conference.

---

> > > ### Author Response · Authors · 2024-08-12
> > >
> > > We sincerely thank you for acknowledging our work. Thanks again for your time and effort in reviewing our work.

---

### Decision · Program_Chairs · 2024-09-25

**Decision:**

Accept (poster)

**Comment:**

This paper is concerned with few shot node classification setting of graph neural networks. The difficulty of such task lies in quick and generalizable adaptation of the model using only few labels. To enhance such generalization, authors propose to incorporate sharpness aware minimization (finding flat local optima in the loss surface) during GNN adaptation. Direct application of SAM is computationally expensive (2x), and authors propose new modifications to SAM. Namely, substitution GNNs with multilayer perceptrons (for loss minimization)  and a setting when perturbations are approximated for few iterations and subsequently recalculated exactly. Provided experiments demonstrate that the method is not only significantly faster than the original SAM but also outperforms it in terms of accuracy.
This paper is clearly and well written and the proposed method was extensively evaluated via carefully crafted experiments and ablations, demonstrating superior performance. Reviewers found that this method has a clear practical application and while not very "novel", it is witty at applying tried and tested technique in a new setting (GNN) in a way that significantly reduces the computational burden.